# On the Overlooked Pitfalls of Weight Decay and How to Mitigate Them: A Gradient-Norm Perspective

Zeke Xie[1,2], Zhiqiang Xu[3], Jingzhao Zhang[4], Issei Sato[1], and Masashi Sugiyama[2,1]

[1]The University of Tokyo
[2]RIKEN Center for AIP
[3]MBZUAI
[4]Tsinghua University
Correspondence: *zekexie16@gmail.com*, *Zhiqiang.Xu@mbzuai.ac.ae*

## Abstract

Weight decay is a simple yet powerful regularization technique that has been very widely used in training of deep neural networks (DNNs). While weight decay has attracted much attention, previous studies fail to discover some overlooked pitfalls on large gradient norms resulted by weight decay. In this paper, we discover that, weight decay can unfortunately lead to large gradient norms at the final phase (or the terminated solution) of training, which often indicates bad convergence and poor generalization. To mitigate the gradient-norm-centered pitfalls, we present the first practical scheduler for weight decay, called the *Scheduled Weight Decay* (SWD) method that can dynamically adjust the weight decay strength according to the gradient norm and significantly penalize large gradient norms during training. Our experiments also support that SWD indeed mitigates large gradient norms and often significantly outperforms the conventional constant weight decay strategy for *Adaptive Moment Estimation* (Adam).

## 1 Introduction

Deep learning [25] has achieved great success in numerous fields over the last decade. Weight decay is the most popular regularization technique for training deep neural networks that generalize well [24]. In deep learning, there exist two types of "weight decay": $L_2$ regularization and weight decay.

People commonly use the first type of $L_2$ regularization for training of deep neural networks as $\theta_t = \theta_{t-1} - \eta_t \nabla(L(\theta_{t-1}) + \lambda_{L_2} \|\theta_{t-1}\|^2)$, where $\nabla$ is the gradient operator, $\| \cdot \|$ is the $L_2$-norm, $t$ is the index of iterations, $\theta$ denotes the model weights, $\eta_t$ is the learning rate, $L(\theta)$ is the training loss, and $\lambda_{L_2}$ is the $L_2$ regularization hyperparameter. In principle, the $L_2$ regularization strength is chosen to be constant during training of DNNs and people do not need to schedule $L_2$ regularization like learning rate decay. The second type of weight decay was first proposed by Hanson and Pratt [13] to directly regularize the weight norm:

$$\theta_t = (1 - \lambda')\theta_{t-1} - \eta_t \frac{\partial L(\theta_{t-1})}{\partial \theta}, \tag{1}$$

where $\lambda'$ is the (vanilla) weight decay hyperparameter.

While the two types of "weight decay" seem similar to Stochastic Gradient Descent (SGD), Loshchilov and Hutter [32] reported that they can make huge differences in adaptive gradient methods, such as Adam [22]. Adam often generalizes worse and finds sharper minima than SGD [40, 56, 46] for popular convolutional neural networks (CNNs), where the flat minima have been argued to be closely related with good generalization [16, 17, 14, 51, 21]. Understanding and bridging the generalization

37th Conference on Neural Information Processing Systems (NeurIPS 2023).

Table 1: Test performance comparison of $L_2$ regularization, Decoupled Weight Decay, and SWD for Adam, which corresponds to Adam, AdamW, and the proposed AdamS, respectively. We report the mean and the standard deviations (as the subscripts) of the optimal test errors. We make the lowest two test errors bold for each model. The SWD method even enables Adam to generalize as well as SGD and even outperform complex Adam variants for the five popular models.

| DATASET | MODEL | ADAMS | ADAM | ADAMW | SGD | AMSGRAD | ADABOUND | PADAM | YOGI | RADAM |
|---|---|---|---|---|---|---|---|---|---|---|
| CIFAR-10 | RESNET18 | $\mathbf{4.91}_{\mathbf{0.04}}$ | $6.96_{0.02}$ | $5.08_{0.07}$ | $\mathbf{5.01}_{\mathbf{0.03}}$ | $6.16_{0.18}$ | $5.65_{0.08}$ | $5.12_{0.04}$ | $5.87_{0.12}$ | $6.01_{0.10}$ |
| | VGG16 | $\mathbf{6.09}_{\mathbf{0.11}}$ | $7.31_{0.25}$ | $6.59_{0.13}$ | $6.42_{0.02}$ | $7.14_{0.14}$ | $6.76_{0.12}$ | $\mathbf{6.15}_{\mathbf{0.06}}$ | $6.90_{0.22}$ | $6.56_{0.04}$ |
| CIFAR-100 | RESNET34 | $\mathbf{21.76}_{\mathbf{0.42}}$ | $27.16_{0.55}$ | $22.99_{0.40}$ | $\mathbf{21.52}_{\mathbf{0.37}}$ | $25.53_{0.19}$ | $22.87_{0.13}$ | $22.72_{0.10}$ | $23.57_{0.12}$ | $24.41_{0.40}$ |
| | DENSENET121 | $\mathbf{20.52}_{\mathbf{0.26}}$ | $25.11_{0.15}$ | $21.55_{0.14}$ | $\mathbf{19.81}_{\mathbf{0.33}}$ | $24.43_{0.09}$ | $22.69_{0.15}$ | $21.10_{0.23}$ | $22.15_{0.36}$ | $22.27_{0.22}$ |
| | GOOGLENET | $\mathbf{21.05}_{\mathbf{0.18}}$ | $26.12_{0.33}$ | $21.29_{0.17}$ | $\mathbf{21.21}_{\mathbf{0.29}}$ | $25.53_{0.17}$ | $23.18_{0.31}$ | $21.82_{0.17}$ | $24.24_{0.16}$ | $22.23_{0.15}$ |

gap between SGD and Adam have been a hot topic recently. Loshchilov and Hutter [32] argued that directly regularizing the weight norm is more helpful for boosting Adam than $L_2$ regularization, and proposed Decoupled Weight Decay as

$$\theta_t = (1 - \eta_t \lambda_W)\theta_{t-1} - \eta_t \frac{\partial L(\theta_{t-1})}{\partial \theta}, \qquad (2)$$

where $\lambda_W$ is the decoupled weight decay hyperparameter. We display Adam with $L_2$ regularization and Adam with Decoupled Weight Decay (AdamW) in Algorithm 1. To avoid abusing notations, we strictly distinguish $L_2$ regularization and weight decay in the following discussion.

While fine analysis of weight decay has attracted much attention recently [52, 39, 4], some pitfalls of weight decay have still been largely overlooked. This paper analyzes the currently overlooked pitfalls of (the second-type) weight decay and studies how to mitigate them by a gradient-norm-aware scheduler. The main contributions can be summarized as follows.

First, supported by theoretical and empirical evidence, we discover that, weight decay can lead to large gradient norms especially at the final phase of training. The large gradient norm problem is significant in the presence of scheduled or adaptive learning rates, such as Adam. From the perspective of optimization, a large gradient norm at the final phase of training often leads to poor convergence. From the perspective of generalization, penalizing the gradient norm is regarded as beneficial regularization and may improve generalization [27, 3, 2, 55]. However, previous studies fail to discover that the pitfalls of weight decay may sometimes significantly hurt convergence and generalization.

Second, to the best of our knowledge, we are the first to demonstrate the effectiveness of scheduled weight decay for mitigating large gradient norms and boosting performance. Inspired by the gradient-norm generalization measure and the stability analysis of stationary points, we show that weight decay should be scheduled with the gradient norm for adaptive gradient methods. We design a gradient-norm-aware scheduler for weight decay, called Scheduled Weight Decay (SWD), which significantly boosts the performance of weight decay and bridges the generalization gap between Adam and SGD. Adam with SWD (AdamS) is displayed in Algorithm 2. The empirical results in Table 1 using various DNNs support that SWD often significantly surpasses unscheduled weight decay for Adam.

**Algorithm 1: Adam/AdamW**

$g_t = \nabla L(\theta_{t-1}) + \lambda\theta_{t-1}$;
$m_t = \beta_1 m_{t-1} + (1 - \beta_1)g_t$;
$v_t = \beta_2 v_{t-1} + (1 - \beta_2)g_t^2$;
$\hat{m}_t = \frac{m_t}{1-\beta_1^t}$;
$\hat{v}_t = \frac{v_t}{1-\beta_2^t}$;
$\theta_t = \theta_{t-1} - \frac{\eta}{\sqrt{\hat{v}_t}+\epsilon}\hat{m}_t - \eta\lambda\theta_{t-1}$;

**Algorithm 2: AdamS**

$g_t = \nabla L(\theta_{t-1})$;
$m_t = \beta_1 m_{t-1} + (1 - \beta_1)g_t$;
$v_t = \beta_2 v_{t-1} + (1 - \beta_2)g_t^2$;
$\hat{m}_t = \frac{m_t}{1-\beta_1^t}$;
$\hat{v}_t = \frac{v_t}{1-\beta_2^t}$;
$\bar{v}_t = mean(\hat{v}_t)$ (gradient-norm-aware);
$\theta_t = \theta_{t-1} - \frac{\eta}{\sqrt{\hat{v}_t}+\epsilon}\hat{m}_t - \frac{\eta}{\sqrt{\bar{v}_t}+\epsilon}\lambda\theta_{t-1}$;

**Structure of this paper.** In Section 2, we analyze the overlooked pitfalls of weight decay, namely large gradient norms. In Section 3, we discuss why large gradient norms are undesired. In Section 4, we propose SWD to mitigate the large gradient norm and boosting performance of Adam. In Section 5, we empirically demonstrate the effectiveness of SWD. In Section 6, we conclude our work.

## 2 Overlooked Pitfalls of Weight Decay

In this section, we report the overlooked pitfalls of weight decay, on convergence, stability, and gradient norms.

**Weight decay should be coupled with the learning rate scheduler.** The vanilla weight decay described by Hanson and Pratt [13] is given by Equation (1). A more popular implementation for vanilla SGD in modern deep learning libraries, such as PyTorch and TensorFlow, is given by Equation (2), where weight decay is coupled with the learning rate scheduler. While existing studies did not formally study why weight decay should be coupled with the learning rate scheduler, this trick has been adopted by most deep learning libraries.

We first take vanilla SGD as a studied example, where no momentum is involved. It is easy to see that vanilla SGD with $L_2$ regularization $\frac{\lambda}{2}\|\theta\|^2$ is also given by Equation (2). Suppose the learning rate is fixed in the whole training procedure. Then Equation (2) will be identical to Equation (1) if we simply choose $\lambda' = \eta\lambda$. However, learning rate decay is quite important for training of deep neural networks. So Equation (2) is not identical to Equation (1) in practice.

**Which implementation is better?** In Figure 1, we empirically verified that the popular Equation-(2)-based weight decay indeed outperforms the vanilla implementation in Equation (1). We argue that Equation-(2)-based weight decay is theoretically better than Equation-(1)-based weight decay in terms of the stability of the stationary points. While we use the notations of *"stable"/"unstable" stationary points* for simplicity, we formally term "stable"/"unstable" stationary points as *strongly/weakly stationary points* in Definition 1. This clarifies that there can exist a gap between true updating and pure gradient-based updating for stationary points.

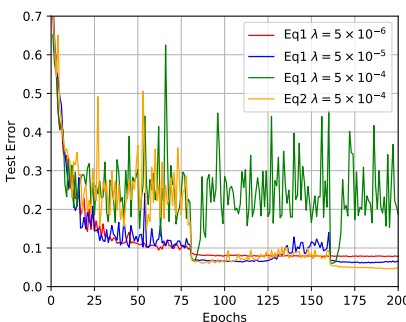

**Definition 1** (Strongly/Weakly Stationary Points). *Suppose $\mathcal{A} : \Theta \rightarrow \Theta$ is a function that updates model parameters during training where $\Theta$ is the parameter space. We define $\theta^\star$ a strongly stationary point for $\mathcal{A}$ if $\|\theta^\star - \mathcal{A}(\theta^\star)\| = 0$ holds during the whole training process. We define $\theta^\star$ a weakly stationary point for $\mathcal{A}$ if $\|\theta^\star - \mathcal{A}(\theta^\star)\| = 0$ holds only for some iterations.*

The training objective at step-$t$ of Equation (1) is $L(\theta) + \frac{\lambda'}{\eta_t}\|\theta\|^2$, while that of Equation (2) is $L(\theta) + \lambda\|\theta\|^2$. Thus, stationary points of the regularized loss given by Equation (2) are stable during training, while that given by Equation (1) is unstable due to the dynamic training objective.

Figure 1: We compared Equation-(1)-based weight decay and Equation-(2)-based weight decay by training ResNet18 on CIFAR-10 via vanilla SGD. In the presence of a popular learning rate scheduler, Equation-(2)-based weight decay shows better test performance. It supports that the form $-\eta_t\lambda\theta$ is a better weight decay implementation than $-\lambda'\theta$.

**Weight decay increases gradient norms at the final phase of deterministic optimization and stochastic optimization.** Following the standard convergence analysis of Gradient Descent (GD) [6] with the extra modification of weight decay, we prove Theorem 1 and show that even GD with vanilla weight decay may not converge to any non-zero stationary point due to missing stability of the stationary points.

**Theorem 1** (Non-convergence due to vanilla weight decay and scheduled learning rates). *Suppose learning dynamics is governed by GD with vanilla weight decay (Equation (1)) and the learning rate $\eta_t \in (0, +\infty)$ holds. If $\exists\delta$ such that satisfies $0 < \delta \leq |\eta_t - \eta_{t+1}|$ for any $t > 0$, then the learning dynamics cannot converge to any non-zero stationary point satisfying the condition*

$$\max(\|\nabla f_t(\theta_t)\|^2, \|\nabla f_t(\theta_{t+1})\|^2) \geq \frac{\lambda'^2\sigma^2\|\theta^\star\|^2}{\eta_t\eta_{t+1}} > 0,$$

*where $f_t(\theta) = L(\theta) + \frac{\lambda'}{\eta_t}\|\theta\|^2$ is the regularized loss.*

We leave the proof in Appendix A.1. Theorem 1 means that even if the gradient norm is zero at the $t$-th step, the gradient norm at the $t + 1$-th step will not be zero. Thus, the solution does not

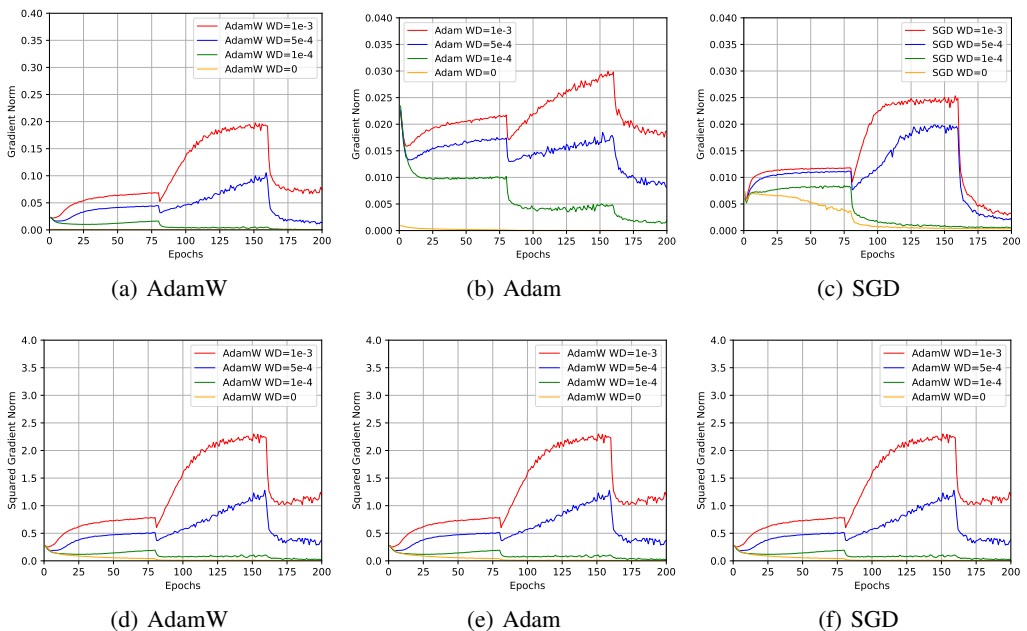

Figure 2: Weight decay increases the gradient norms and the squared gradient norms ResNet18 for various optimizers. Top Row: the gradient norms. Bottom Row: the squared gradient norms.

converge to any non-zero stationary point. GD with unstable minima has no theoretical convergence guarantee like GD [41]. More importantly, due to unstable stationary points, the gradient norms (of the regularized/original loss) naturally also do not converge to zero. This may explain why weight decay should be coupled with the learning rate to improve stability of stationary points during training. This non-convergence problem is especially significant when $\delta$ is relatively large in the presence of scheduled or adaptive learning rates, such as Adam.

Existing studies [48, 12] proved that the gradient norm bound of SGD in convergence analysis depends on the properties of the training objective function and gradient noise. We further prove Theorem 2 and reveal how the gradient norm bound in convergence analysis depends on the weight decay strength $\lambda$ when weight decay is involved. We leave the proof in Appendix A.2.

**Theorem 2** (Convergence analysis on weight decay). *Assume that $L(\theta)$ is an $\mathcal{L}$-smooth function[1], $L$ is lower-bounded as $L(\theta) \geq L^\star$, $L(\theta, X)$ is the loss over one minibatch $X$, $\mathbb{E}[\nabla L(\theta, X) - \nabla L(\theta)] = 0$, $\mathbb{E}[\|\nabla L(\theta, X) - \nabla L(\theta)\|^2] \leq \delta^2$, and $\|\nabla L(\theta)\| \leq G$ for any $\theta$. Let SGD optimize $f(\theta) = L(\theta) + \frac{\lambda}{2}\|\theta\|^2$ for $t + 1$ iterations. If $\eta \leq \frac{C}{\sqrt{t+1}}$, we have*

$$\min_{k=0,\ldots,t} \mathbb{E}[\|\nabla f(\theta_k)\|^2] \leq \frac{1}{\sqrt{t+1}} [C_1 + C_2],\qquad(3)$$

*where*

$$C_1 = \frac{L(\theta_0) + \frac{\lambda}{2}\|\theta_0\|^2 - L^\star}{C},\qquad(4)$$

$$C_2 = C(\mathcal{L} + \lambda)((G + \sup(\lambda\|\theta\|))^2 + \sigma^2),\qquad(5)$$

*and $\sup(\|\theta\|) = \sup_{k=0,\ldots,t} \|\theta_k\|$ is the maximum $L_2$ norm of $\theta$ over the iterations.*

Theorem 1 shows the non-convergence problem with improper weight decay. Theorem 2 further shows that, even with proper convergence guarantees and stable minima, the gradient norm upper bound at convergence still monotonically increases with the weight decay strength $\lambda$. Obviously, Theorem 2 theoretically supports that weight decay can result in large gradient norms. The phenomenon can also be widely observed in the following empirically analysis.

---

[1]It means that $\|\nabla L(\theta_a) - \nabla L(\theta_b)\| \leq \mathcal{L}\|\theta_a - \theta_b\|$ holds for any $\theta_a$ and $\theta_b$.

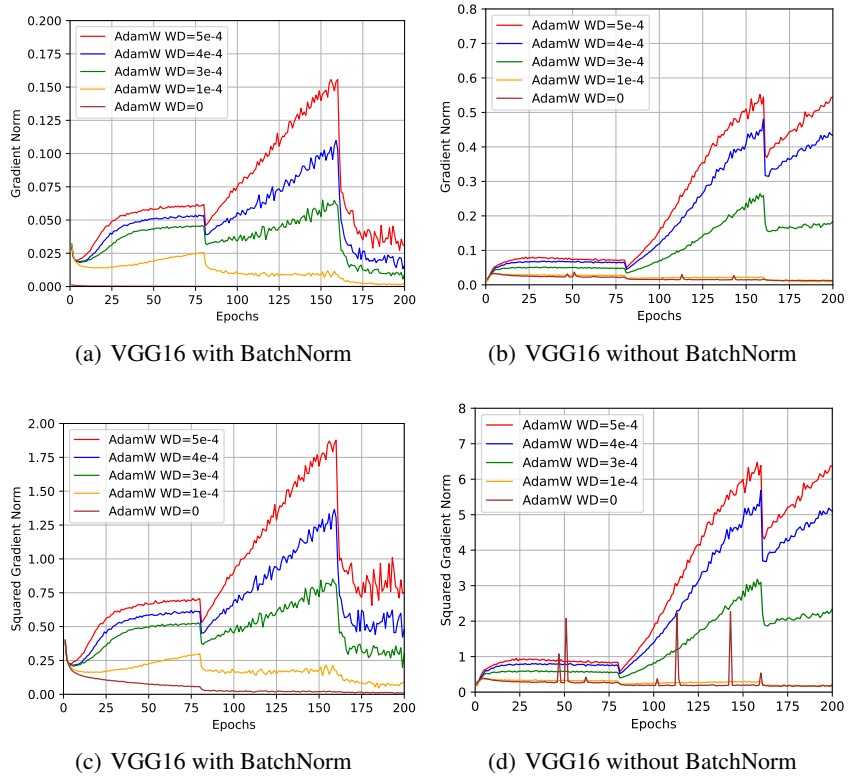

(a) VGG16 with BatchNorm  (b) VGG16 without BatchNorm

(c) VGG16 with BatchNorm  (d) VGG16 without BatchNorm

Figure 3: Weight decay increases the gradient norms and the squared gradient norms of VGG16 with BatchNorm (scale-invariant) and VGG16 without BatchNorm (scale-variant) on CIFAR-10. Top Row: the gradient norms. Bottom Row: the squared gradient norms.

**Empirical evidence of large gradient norms resulted by weight decay.** Theorem 1 suggests that both unstability and strength of weight decay may increase the gradient norm even in full-batch training. While weight decay is helpful for regularizing the model weights' norm, we clearly observed in Figure 2 that weight decay significantly increases the gradient norms in training of DNNs as we expect. Note that the displayed weight decay of AdamW is rescaled by the factor $0.001$. The observation generally holds for various optimizers, such as AdamW, Adam and SGD. Figure 3 further shows that the observation holds for both scale-invariant models (VGG16 with BatchNorm) and scale-variant models (VGG16 without BatchNorm). Moreover, the observation still holds for gradient norms with or without regularization included, because the numerical difference due to regularization is smaller than their gradient norms by multiple orders of magnitude in the experiments. Zhang et al. [52] argued that weight decay increases the "effective learning rate" due to smaller weight norms resulted by weight decay. However, the "effective learning rate" interpretation may not explain large gradient norms, because a small weight norm does not necessarily indicate a large gradient norm. Interestingly, Figure 3 shows that the gradient curves of VGG16 without BatchNorm and weight decay sometimes exhibit strange spikes, which can be wiped out by weight decay.

## 3 Large Gradient Norms Are Undesired

In this section, we demonstrate that the large gradient norm is a highly undesirable property in deep learning because large gradient norms correspond to multiple pitfalls.

First, small gradient norms are required for good convergence, which is desired in optimization theory [36]. Popular optimizers also expect clear convergence guarantees [8]. The optimizers with no convergence guarantees may cause undesirable training behaviors in practice.

Second, regularizing the gradient norm has been recognized as an important beneficial regularization effect in recent papers [28, 3, 11, 55, 54]. Li et al. [28] and Geiping et al. [11] expressed implicit regularization of stochastic gradients as

$$R_{\text{sg}} = \frac{\eta}{4|\mathcal{B}|} \sum_{X \in \mathcal{B}} \|\nabla L(\theta, X)\|^2 = \frac{\eta}{4} \mathcal{G}(\theta), \tag{6}$$

where $L(\theta, X)$ indicate the loss over one data minibatch $X \in \mathcal{B}$, $|\mathcal{B}|$ is the number of minibatches per epoch, and $\mathcal{G}$ is the expected squared gradient norms over each minibatch. Recent studies, Zhao et al. [55] and Zhang et al. [54], also successfully improved generalization by explicitly adding the gradient-norm penalty into stochastic optimization. Obviously, penalizing the gradient norm is considered as an important and beneficial regularizer in deep learning.

Third, large gradient norms may theoretically indicate poor generalization according to the gradient-norm generalization measures [27, 2]. For example, Theorem 11 of Li et al. [27] derived a gradient-norm-based generalization error upper bound that uses the squared gradient norms, written as

$$\text{Sup Gen} = \mathcal{O}\left( \frac{L^\star}{n} \sqrt{\mathbb{E}\left[ \frac{\eta_t^2}{\sigma_t^2} \mathcal{G}(\theta_t) \right]} \right), \tag{7}$$

where $n$ is the training data size, $\sigma_t^2$ is the gradient noise variance at the $t$-th iteration, and $\mathcal{G}(\theta_t) = \frac{1}{|\mathcal{B}|} \sum_{X \in \mathcal{B}} \|\nabla L(\theta, X)\|^2$ is the squared gradient norm with the batch size as 1. An et al. [2] presented more empirical evidences for supporting the gradient-norm-based generalization bound (7). Moreover, large gradient norms are also believed to be closely related to minima sharpness [20, 57, 43, 47, 42, 46, 7], while sharp minima often indicate poor generalization [16, 17, 14, 51, 21]. As our analysis focuses on the large gradient norm at the final phase (or the terminated solution) of training, our conclusion is actually not contradicted to the viewpoint [1, 29, 10] that the gradient-noise regularization effect of large initial learning rates at the early phase of training is beneficial.

We also report that the gradient norm and the squared gradient norm show very similar tendencies in our experiments. For simplicity, we mainly use $\mathcal{G}$ as the default gradient norm measure in this paper, unless we specify it otherwise. As we discussed above, the squared gradient norm $\mathcal{G}$ is more closely related to implicit gradient regularization, generalization bounds, and SWD than other gradient norms. Overall, the theoretical results above supported that the small gradient norm is a desirable property in deep learning. However, our theoretical analysis suggests that weight decay can lead to large gradient norms and hence hurt generalization. We suggest that gradient-norm-centered poor convergence and generalization can lead to the serious but overlooked performance degradation of weight decay.

## 4 Gradient-Norm-Aware Scheduled Weight Decay

In this section, we design the first practical scheduler to boost weight decay by mitigating large gradient norm.

We focus on scheduled weight decay for Adam in this paper for three reasons. First, Adam is the most popular optimization for accelerating training of DNNs. Second, as the adaptivity of Adam updates the learning rate every iteration, Adam suffers more than SGD from the pitfalls of weight decay due to adaptive learning rates. Third, SGD usually employs $L_2$ regularization rather than weight decay.

Adam with decoupled weight decay is often called AdamW [32], which can be written as

$$\theta_t = (1 - \eta\lambda)\theta_{t-1} - \eta v_t^{-\frac{1}{2}} m_t, \tag{8}$$

where $v_t$ and $m_t$ are the exponential moving average of the squared gradients and gradients in Algorithm 1, respectively, and the power notation of a vector means the element-wise power of the vector. We interpret $\eta v_t^{-\frac{1}{2}}$ as the effective learning rate for multiplying the gradients. However, we clearly see that decoupled weight decay couples weight decay with the vanilla learning rate rather than the effective learning rate. Due to the differences of weight decay and $L_2$ regularization, AdamW and Adam optimize different training objectives. The minimum $\hat{\theta}^\star$ of the regularized loss function optimized by AdamW at the $t$-th step is not constant during training. Thus, the regularized loss function optimized by AdamW has unstable minima, which may result in large gradient norms

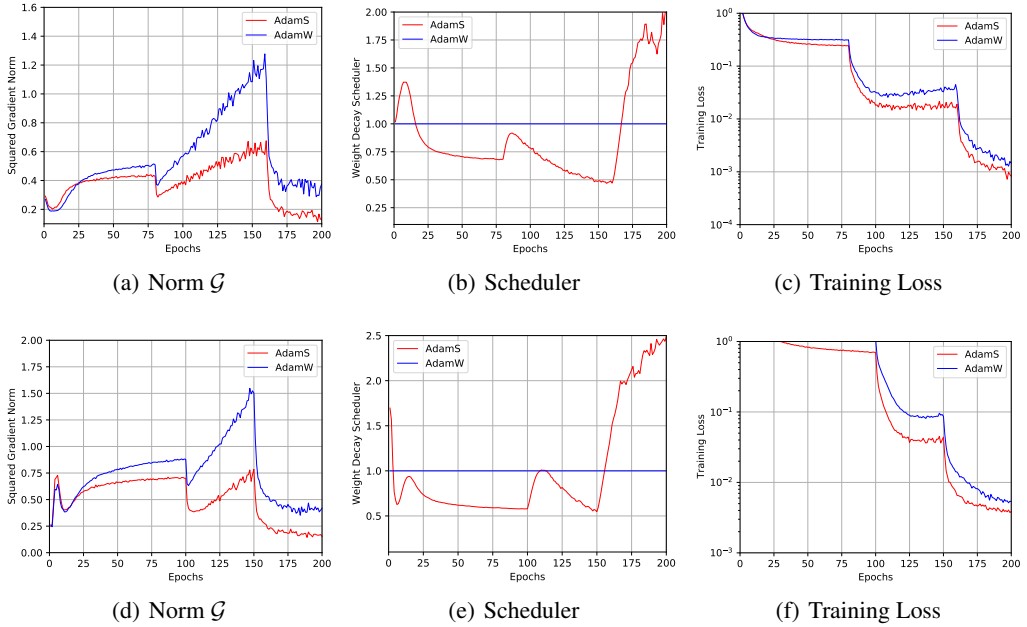

Figure 4: The curves of the weight decay scheduler, $\mathcal{G}$, and training losses of SWD and constant weight decay. Top Row: ResNet18 on CIFAR-10. Bottom Row: ResNet34 on CIFAR-100. SWD mitigates large gradient norms.

suggested by Theorem 1. This may be an internal fault of all optimizers that use adaptive learning rates and weight decay at the same time.

To mitigate the pitfall of large gradient norms, we propose a gradient-norm-aware scheduler for weight decay as

$$\theta_t = (\mathbf{1} - \eta \bar{v}_t^{-\frac{1}{2}} \lambda) \theta_{t-1} - \eta v_t^{-\frac{1}{2}} m_t, \tag{9}$$

where $\bar{v}_t$ is the mean of all elements of the vector $v_t$. Moreover, as $v_t$ is the exponential moving average of squared gradient norms, the factor $\bar{v}_t$ is expected to be equivalent to the squared gradient norms divided by model dimensionality. While the proposed scheduler cannot give perfectly stable minima for Adam, it can penalize large gradient norms by dynamically adjusting weight decay. We call weight decay with this gradient-norm-aware scheduler Scheduled Weight Decay (SWD). The pseudocode is displayed in Algorithm 2. The code is publicly available at GitHub. In Section 5, we empirically demonstrate the advantage of the gradient-norm-aware scheduler. Note that AdamS is slightly slower than AdamW but slightly faster than Adam, while the cost difference is nearly ignorable (usually less than $5\%$).

We note that $\bar{v}$ is not expected to be zero at minima due to stochastic gradient noise, because the variance of the stochastic gradient is directly observed to be much larger than the expectation of the stochastic gradient at/near minima and depends on the Hessian [43]. In some cases, such as full-batch training, it is fine to add a small value (e.g., $10^{-8}$) to $\bar{v}$ to avoid being zero as a divisor.

Lewkowycz and Gur-Ari [26] recently proposed a simple weight decay scheduling method which has a limited practical value because it only works well with a constant learning rate, as the original paper claimed. Bjorck et al. [4] empirically studied early effects of weight decay but did not touch mitigating the pitfalls of weight decay. In contrast, our experiments suggest that SWD indeed effectively penalizes the large gradient norm and hence improve convergence and generalization.

Figure 4 verifies multiple advantages of SWD. First, SWD dynamically adjusts the weight decay strength inverse to $\mathcal{G}$ as we expect. Second, obviously, SWD can effectively penalize large gradient norms compared with constant weight decay during not only the final training phase but also the nearly whole training procedure. Third, SWD can often lead to lower training losses than constant weight decay. Figure 5 clearly reveals that, to choose proper weight decay strength, SWD dynamically

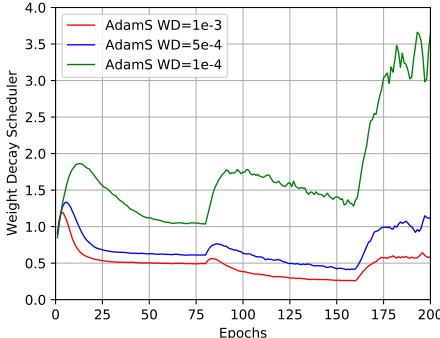 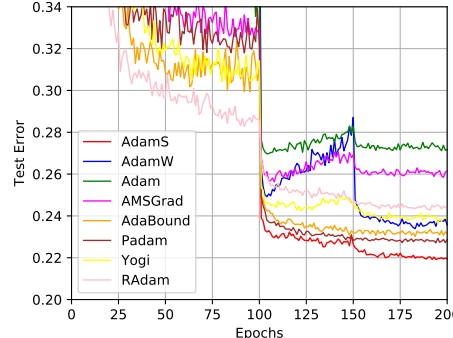

Figure 5: SWD adjusts the scheduler strength according to the initial weight decay hyperparameter. The optimal choice is $\lambda = 5 \times 10^4$. Model: ResNet18. Dataset: CIFAR-10.

Figure 6: The learning curves of all adaptive gradient methods by training ResNet34 on CIFAR-100. AdamS outperforms other Adam variants.

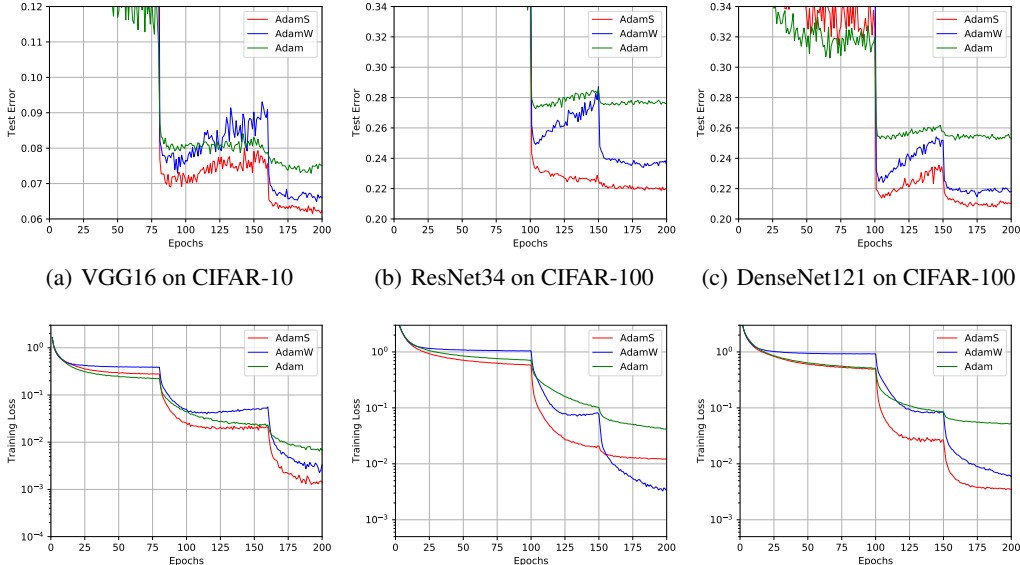

(a) VGG16 on CIFAR-10      (b) ResNet34 on CIFAR-100      (c) DenseNet121 on CIFAR-100

Figure 7: The learning curves of AdamS, AdamW, and Adam on CIFAR-10 and CIFAR-100. AdamS shows significantly better generalization than AdamW and Adam. Top Row: Test curves. Bottom Row: Training curves.

employs a stronger scheduler when the initially selected weight decay hyperparameter $\lambda$ is smaller and a weaker scheduler when $\lambda$ is larger compared to the optimal weight decay choice, respectively.

## 5 Empirical Analysis

In this section, we first empirically verify that weight decay indeed affects gradient norms and further conducted comprehensive experiments to demonstrate the effectiveness of the SWD scheduler in mitigating large gradient norms and boosting performance.

**Models and Datasets.** We train various neural networks, including ResNet18/34/50 [15], VGG16 [37], DenseNet121 [19], GoogLeNet [38], and Long Short-Term Memory (LSTM) [18], on CIFAR-10/CIFAR-100 [23], ImageNet[9], and Penn TreeBank [34]. The optimizers include popular Adam variants, including AMSGrad [35], Yogi [49], AdaBound [33], Padam [5], and RAdam [30]. See Appendix B and C for more details and results.

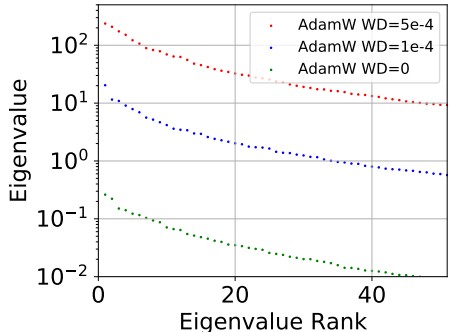
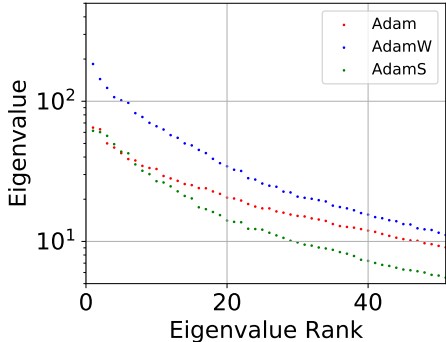

Figure 8: Weight decay significantly increase the minima sharpness in terms of top Hessian eigenvalues. Model: ResNet18. Dataset: CIFAR-10.

Figure 9: Top 50 Hessian eigenvalues of ResNet18 at the minima learned by Adam, AdamW, and AdamS. AdamS learns flatter minima than AdamW.

**Generalization.** Figure 7 shows the learning curves of AdamS, AdamW, and Adam on several benchmarks. In our experiments, AdamS always leads to lower test errors. Figure 19 in Appendix shows that, even if with similar or higher training losses, AdamS still generalizes significantly better than AdamW, Adam, and recent Adam variants. Figure 6 displays the learning curves of all adaptive gradient methods. The test performance of other models can be found in Table 1. The results on ImageNet (See Figure 18 in Appendix) shows that, AdamS can also make improvements over AdamW. Simply scheduling weight decay for Adam by SWD even outperforms complex Adam variants. We report that most Adam variants surprisingly generalize worse than SGD (See Appendix C).

**Minima sharpness.** Previous studies [20, 57, 43, 45, 44, 7] reported that the minima sharpness measured by the Hessian positively correlate to the gradient norm. Note that a number of popular minima sharpness depends on the Hessian at minima [21]. In Figures 8 and 9, we visualize the top Hessian eigenvalues to measure the minima sharpness. Figure 8 shows that stronger weight decay leads to sharper minima. Figure 9 suggests that AdamS learns significantly flatter minima than AdamW. Both large gradient norms and sharp minima correspond to high-complexity solutions and often hurts generalization [16]. This is a surprising finding for weight decay given the conventional belief that weight decay as a regularization technique should encourage low-complexity solutions.

**The Robustness of SWD.** Figure 11 shows that AdamS is more robust to the learning rate and weight decay than AdamW and Adam. AdamS has a much deeper and wider basin than AdamW and Adam. Figure 10 further demonstrates that AdamS consistently outperforms Adam and AdamW under various weight decay hyperparameters. According to Figure 10, we also notice that the optimal decoupled weight decay hyperparameter in AdamW can be very different from $L_2$ regularization and SWD. Thus, AdamW requires re-tuning the weight decay hyperparameter in practice, which is time-consuming. The robustness of selecting hyperparameters can be supported by Figure 5, where the proposed scheduler also depends on the initial weight decay strength. It means that the proposed gradient-norm-aware scheduler exhibit the behavior of negative feedback to stabilize the strength of weight decay during training.

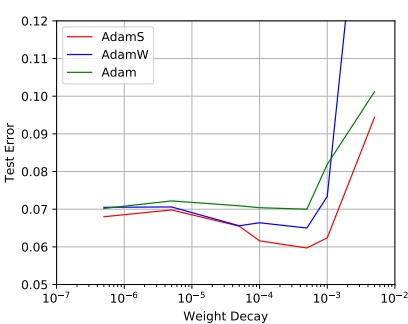

Figure 10: The test errors of VGG16 on CIFAR-10 with various weight decay rates. The displayed weight decay value of AdamW has been rescaled by the factor = 0.001. A similar experimental result for ResNet34 is presented in Appendix C.

**Cosine learning rate schedulers and warm restarts.** We conducted comparative experiments on AdamS, AdamW, and Adam in the presence of cosine annealing schedulers and warm restarts proposed by Loshchilov and Hutter [31]. We set the learning rate scheduler with a recommended

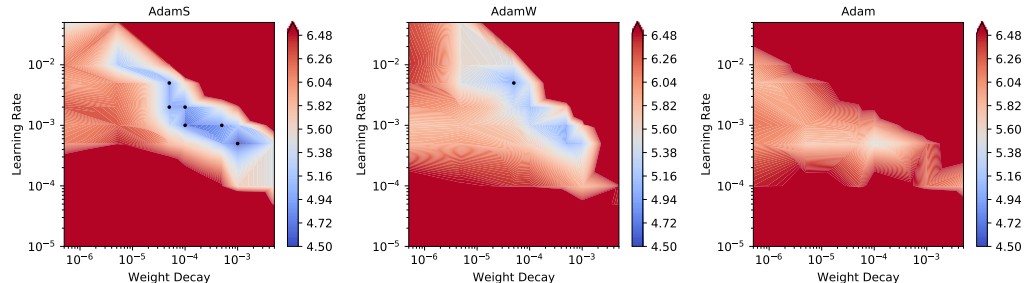

Figure 11: The test errors of ResNet18 on CIFAR-10. AdamS has a much deeper and wider basin near dark points ($\leq 4.9\%$). The optimal test errors of AdamS, AdamW, and Adam are $4.52\%$, $4.90\%$, and $5.49\%$, respectively. The displayed weight decay of AdamW has been rescaled by the factor $= 0.001$.

setting of Loshchilov and Hutter [31]. Our experimental results in Figure 12 suggest that AdamS consistently outperforms AdamW and Adam in terms of both training losses and test errors. It demonstrates that, with complex learning rate schedulers, the advantage of SWD still holds.

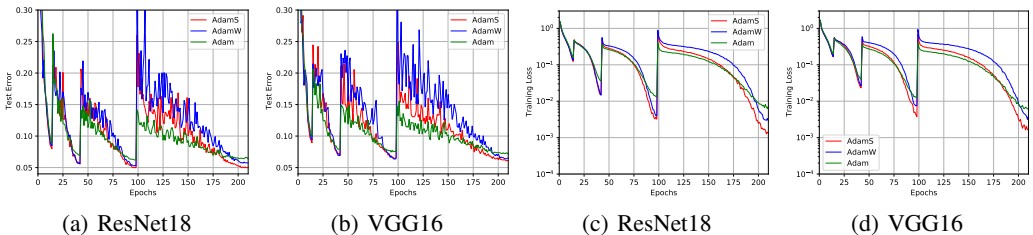

| (a) ResNet18 | (b) VGG16 | (c) ResNet18 | (d) VGG16 |

Figure 12: The test curves and training curves of ResNet18 and VGG16 on CIFAR-10 with cosine annealing and warm restart schedulers. The weight decay hyperparameter: $\lambda_{L_2} = \lambda_S = 0.0005$ and $\lambda_W = 0.5$. Left Two Figures: Test curves. Right Two Figures: Training curves. AdamS yields significantly lower test errors and training losses than AdamW and Adam.

**Limitations.** While the proposed scheduler boosts weight decay in most experiments, the improvement of SWD may be limited sometimes. SWD does not work well particularly when $L_2$ regularization even yields better test results than weight decay (See the experiments on Language Modeling Figure 13 in Appendix C). Our paper does not touch the pitfall of $L_2$ regularization and how to schedule its strength. We leave scheduling $L_2$ regularization as future work.

# 6 Conclusion

While weight decay has attracted much attention, previous studies failed to recognize the problems of unstable stationary points and large gradient norm caused by weight decay. In this paper, we theoretically and empirically studied the overlooked pitfalls of weight decay from the gradient-norm perspective. To mitigate the pitfalls, we propose a simple but effective gradient-norm-aware scheduler for weight decay. Our empirical results demonstrate that SWD can effectively penalize large gradient norms and improve convergence. Moreover, SWD often significantly improve generalization over constant schedulers. The generalization gap between SGD and Adam can be almost closed by such a simple weight decay scheduler for CNNs.

We emphasize that the proposed method is the first scheduled weight decay method rather than a novel optimizer. Although our analysis mainly focused on Adam, SWD can be easily combined with other optimizers, including SGD and Adam variants. To the best of our knowledge, this is the first formal touch on the art of scheduling weight decay to boost weight decay. We believe that scheduled weight decay as a new line of research may inspire more theories and algorithms that will help further understanding and boosting weight decay in future. Our work just made the first step along this line.

## Acknowledgement

MS was supported by JST CREST Grant Number JPMJCR18A2.

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

# A Proofs

## A.1 Proof of Theorem 1

*Proof.* We first write the regularized loss function corresponding to SGD with vanilla weight decay as

$$f_t(\theta) = L(\theta) + \frac{\lambda'}{2\eta_t}\|\theta\|^2 \tag{10}$$

at $t$-th step.

If the corresponding $L_2$ regularization $\frac{\lambda'}{2\eta_t}$ is unstable during training, the regularized loss function $f_t(\theta)$ will also be a time-dependent function and has no non-zero stable stationary points.

Suppose we have a non-zero solution $\theta^\star$ which is a stationary point of $f(\theta, t)$ at $t$-th step and SGD finds $\theta_t = \theta^\star$ at $t$-th step.

Even if the gradient of $f_t(\theta)$ at $t$-step is zero, we have the gradient at $(t+1)$-th step as

$$g_{t+1} = \nabla f_{t+1}(\theta^\star) = \lambda'(\eta_t^{-1} - \eta_{t+1}^{-1})\theta^\star. \tag{11}$$

It means that

$$\|g_{t+1}\|^2 = \lambda'^2(\eta_t^{-1} - \eta_{t+1}^{-1})^2\|\theta^\star\|^2 \geq \frac{\lambda'^2\delta^2\|\theta^\star\|^2}{\eta_t\eta_{t+1}} \tag{12}$$

To achieve convergence, we must have $\|g_{t+1}\|^2 = 0$.

It requires $(\eta_t^{-1} - \eta_{t+1}^{-1})^2 = 0$ or $\|\theta^\star\|^2 = 0$.

Theorem 2.2 of Shapiro and Wardi [36] told us that the learning rate should be small enough for convergence. Obviously, we have $\eta < \infty$ in practice.

As $\eta_t = \eta_{t+1}$ does not hold, SGD cannot converging to any non-zero stationary point.

The proof is now complete. $\qquad\square$

## A.2 Proof of Theorem 2

Before formally completing the proof, we first introduce a useful Lemma 1, which is a specialized case of Theorem 1 in Yan et al. [48] with $\beta = s = 0$.

**Lemma 1** (Convergence of SGD). *Assume that $L(\theta)$ is an $\mathcal{L}$-smooth function[2], $L$ is lower bounded as $L(\theta) \geq L^\star$, $\mathbb{E}[\nabla L(\theta, X) - \nabla L(\theta)] = 0$, $\mathbb{E}[\|\nabla L(\theta, X) - \nabla L(\theta)\|^2] \leq \delta^2$, $\|\nabla L(\theta)\| \leq G$ for any $\theta$. Let SGD optimize $L$ for $t + 1$ iterations. If $\eta \leq \frac{C}{\sqrt{t+1}}$, we have*

$$\min_{k=0,\dots,t} \mathbb{E}[\|\nabla L(\theta_k)\|^2] \leq \frac{C_0}{\sqrt{t+1}}, \tag{13}$$

*where $C_0 = \left[\frac{L(\theta_0) - L^\star}{C} + C\mathcal{L}(G^2 + \sigma^2)\right]$.*

*Proof.* Given the conditions of $L(\theta)$ in Lemma 1, we may obtain the resulted conditions of $f(\theta) = L(\theta) + \frac{\lambda}{2}\|\theta\|^2$.

As $L(\theta)$ is an $\mathcal{L}$-smooth function, we have

$$\|\nabla f(\theta_a) - \nabla f(\theta_b)\| = \|\nabla L(\theta_a) - \nabla L(\theta_b) + \lambda(\theta_a - \theta_b)\| \leq (\mathcal{L} + \lambda)\|\theta_a - \theta_b\| \tag{14}$$

holds for any $\theta_a$ and $\theta_b$. It shows that $f(\theta)$ is an $(\mathcal{L} + \lambda)$-smooth function.

As $L$ is lower bounded as $L(\theta) \geq L^\star$, we have

$$f^\star \geq L^\star. \tag{15}$$

---

[2]It means that $\|\nabla L(\theta_a) - \nabla L(\theta_b)\| \leq \mathcal{L}\|\theta_a - \theta_b\|$ holds for any $\theta_a$ and $\theta_b$.

As $\mathbb{E}[\nabla L(\theta, X) - \nabla L(\theta)] = 0$, we have

$$\mathbb{E}[\nabla f(\theta, X) - \nabla f(\theta)] = \mathbb{E}[\nabla L(\theta, X) - \nabla L(\theta)] = 0. \tag{16}$$

As $\mathbb{E}[\|\nabla L(\theta, X) - \nabla L(\theta)\|^2] \leq \delta^2$, we have

$$\mathbb{E}[\|\nabla f(\theta, X) - \nabla f(\theta)\|^2] = \mathbb{E}[\|\nabla L(\theta, X) - \nabla L(\theta)\|^2] \leq \delta^2. \tag{17}$$

As $\|\nabla L(\theta)\| \leq G$, we have

$$\|\nabla f(\theta)\| = \|\nabla L(\theta) + \lambda\theta\| \leq G + \lambda\|\theta\|_{\max}, \tag{18}$$

where $\|\theta\|_{\max}$ is the maximum $L_2$ norm of any $\theta$.

Introducing the derived conditions Eq. (12) - (16) for $f$ into Lemma 1, we may treat $f$ as the objective optimized by SGD. Then we have

$$\min_{k=0,\ldots,t} \mathbb{E}[\|\nabla f(\theta_k)\|^2] \leq \frac{1}{\sqrt{t+1}} \left[ \frac{f(\theta_0) - f^\star}{C} + C(\mathcal{L} + \lambda)((G + \lambda\|\theta\|_{\max})^2 + \sigma^2) \right] \tag{19}$$

$$\leq \frac{1}{\sqrt{t+1}} \left[ \frac{L(\theta_0) + \frac{\lambda}{2}\|\theta_0\|^2 - L^\star}{C} + C(\mathcal{L} + \lambda)((G + \lambda\|\theta\|_{\max})^2 + \sigma^2) \right] \tag{20}$$

Obviously, the gradient norm upper bound in convergence analysis monotonically increases as the weight decay strength $\lambda$.

The proof is complete.

$\square$

## B    Experimental Details

**Computational environment.** The experiments are conducted on a computing cluster with GPUs of NVIDIA® Tesla™ P100 16GB and CPUs of Intel® Xeon® CPU E5-2640 v3 @ 2.60GHz.

### B.1    Image Classification on CIFAR-10 and CIFAR-100

**Data Preprocessing For CIFAR-10 and CIFAR-100:** We perform the common per-pixel zero-mean unit-variance normalization, horizontal random flip, and $32 \times 32$ random crops after padding with 4 pixels on each side.

**Hyperparameter Settings:** We select the optimal learning rate for each experiment from $\{0.0001, 0.001, 0.01, 0.1, 1, 10\}$ for non-adaptive gradient methods. We use the default learning rate for adaptive gradient methods in the experiments of Table 1, while we also compared Adam, AdamW, AdamS under various learning rates and batch sizes in other experiments. In the experiments on CIFAF-10 and CIFAR-100: $\eta = 0.1$ for SGD and SGDS; $\eta = 0.001$ for Adam, AdamW, AdamS, AMSGrad, Yogi, AdaBound, and RAdam; $\eta = 0.01$ for Padam. For the learning rate schedule, the learning rate is divided by 10 at the epoch of $\{80, 160\}$ for CIFAR-10 and $\{100, 150\}$ for CIFAR-100, respectively. The batch size is set to 128 for both CIFAR-10 and CIFAR-100.

The strength of $L_2$ regularization and SWD is default to $0.0005$ as the baseline. Considering the linear scaling rule, we choose $\lambda_W = \frac{\lambda_{L_2}}{\eta}$. Thus, the weight decay of AdamW uses $\lambda_W = 0.5$ for CIFAR-10 and CIFAR-100. The basic principle of choosing weight decay strength is to let all optimizers have similar convergence speed.

We set the momentum hyperparameter $\beta_1 = 0.9$ for SGD and SGDS. As for other optimizer hyperparameters, we apply the default hyperparameter settings directly.

We repeated each experiment for three times in the presence of the error bars.

We leave the empirical results with the weight decay setting $\lambda = 0.0001$ in Appendix C.

Table 2: Test performance comparison of optimizers with $\lambda_{L_2} = \lambda_S = 0.0001$ and $\lambda_W = 0.1$, which is a common weight decay setting in related papers. AdamS still show better test performance than popular adaptive gradient methods and SGD.

| DATASET | MODEL | SGD | ADAMS | ADAM | AMSGRAD | ADAMW | ADABOUND | PADAM | YOGI | RADAM |
|---------|-------|-----|-------|------|---------|-------|----------|-------|------|-------|
| CIFAR-10 | RESNET18 | 5.58 | **4.69** | 6.08 | 5.72 | 5.33 | 6.87 | 5.83 | 5.43 | 5.81 |
| | VGG16 | 6.92 | **6.16** | 7.04 | 6.68 | 6.45 | 7.33 | 6.74 | 6.69 | 6.73 |
| CIFAR-100 | RESNET34 | 24.92 | **23.50** | 25.56 | 24.74 | 23.61 | 25.67 | 25.39 | 23.72 | 25.65 |
| | DENSENET121 | **20.98** | 21.35 | 24.39 | 22.80 | 22.23 | 24.23 | 22.26 | 22.40 | 22.40 |
| | GOOGLENET | 21.89 | **21.60** | 24.60 | 24.05 | 21.71 | 25.03 | 26.69 | 22.56 | 22.35 |

## B.2 Image classification on ImageNet

**Data Preprocessing For ImageNet:** For ImageNet, we perform the per-pixel zero-mean unit-variance normalization, horizontal random flip, and the resized random crops where the random size (of 0.08 to 1.0) of the original size and a random aspect ratio (of $\frac{3}{4}$ to $\frac{4}{3}$) of the original aspect ratio is made.

**Hyperparameter Settings for ImageNet:** We select the optimal learning rate for each experiment from $\{0.0001, 0.001, 0.01, 0.1, 1, 10\}$ for all tested optimizers. For the learning rate schedule, the learning rate is divided by 10 at the epoch of $\{30, 60\}$. We train each model for 90 epochs. The batch size is set to 256. The weight decay hyperparameter of AdamS, AdamW, Adam are chosen from $\{5 \times 10^{-6}, 5 \times 10^{-5}, 5 \times 10^{-4}, 5 \times 10^{-3}, 5 \times 10^{-2}\}$. As for other optimizer hyperparameters, we still apply the default hyperparameter settings directly.

## B.3 Language Modeling

We use a classical language model, Long Short-Term Memory (LSTM) [18] with 2 layers, 512 embedding dimensions, and 512 hidden dimensions, which has 14 million model parameters and is similar to the "medium LSTM" in Zaremba et al. [50]. Note that our baseline performance is better than the reported baseline performance in Zaremba et al. [50]. The benchmark task is the word-level Penn TreeBank [34]. We empirically compared AdamS, AdamW, and Adam under the common and same conditions.

**Hyperparameter Settings.** Batch Size: $B = 20$. BPTT Size: $bptt = 35$. Learning Rate: $\eta = 0.001$. We run the experiments under various weight decay selected from $\{10^{-4}, 5 \times 10^{-5}, 10^{-5}, 5 \times 10^{-6}, 10^{-6}, 5 \times 10^{-7}, 10^{-7}\}$. The dropout probability is set to 0.5. We clipped gradient norm to 1.

## C   Supplementary Figures and Results of Adaptive Gradient Methods

**Popular Adam variants often generalize worse than SGD.** A few Adam variants tried to fix the hidden problems in adaptive gradient methods, including AdamW Loshchilov and Hutter [32], AMSGrad [35] and Yogi [49]. A recent line of research, such as AdaBound [33], Padam [5], and RAdam [30], believes controlling the adaptivity of learning rates may improve generalization. This line of research usually introduces extra hyperparameters to control the adaptivity, which requires more efforts in tuning hyperparameters. However, we and Zhang et al. [53] found that this argument is contradicted with our comparative experimental results (see Table 1). In our empirical analysis, most advanced Adam variants may narrow but not completely close the generalization gap between adaptive gradient methods and SGD. SGD with a fair weight decay hyperparameter as the baseline performance usually generalizes better than recent adaptive gradient methods. The main problem may lie in weight decay. SGD with weight decay $\lambda = 0.0001$, a common setting in related papers, is often not a good baseline, as $\lambda = 0.0005$ often shows better generalization on CIFAR-10 and CIFAR-100. We also conduct comparative experiments with $\lambda = 0.0001$. Under the setting $\lambda = 0.0001$, while some existing Adam variants may outperform SGD sometimes due to the lower baseline performance of SGD, AdamS shows superior test performance. For example, for ResNet18 on CIFAR-10, the test error of AdamS is lower than SGD by nearly one point and no other Adam variant may compare with AdamS.

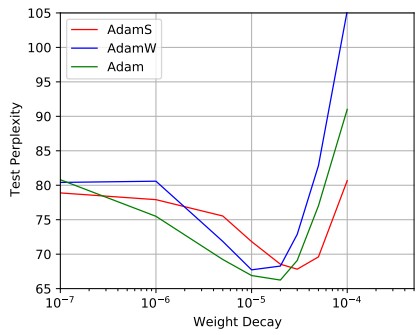

Figure 13: Language modeling under various weight decay. Note that the lower perplexity is better.

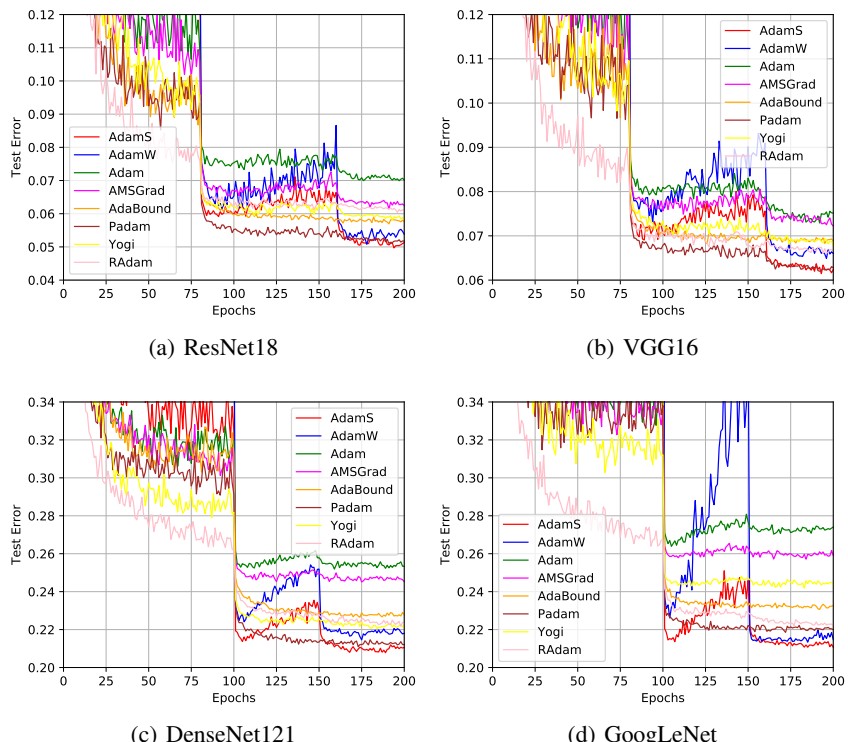

(a) ResNet18

(b) VGG16

(c) DenseNet121

(d) GoogLeNet

Figure 14: The learning curves of adaptive gradient methods.

**Language Modeling.** It is well-known that, different from computer vision tasks, the standard Adam (with $L_2$ regularization) is the most popular optimizer for language models. Figure 13 in Appendix C demonstrates that the conventional belief is true that the standard $L_2$ regularization yields better test results than both Decoupled Weight Decay and SWD. The weight decay scheduler suitable for language models is an open problem.

We report the learning curves of all adaptive gradient methods in Figure 14. They shows that vanilla Adam with SWD can outperform other complex variants of Adam.

Figure 15 displays the scatter plot of training losses and test errors during final 40 epochs of training DenseNet121 on CIFAR-100.

Figure 16 displays the test performance of AdamS, AdamW, and Adam under various weight decay hyperparameters of ResNet34 on CIFAR-100.

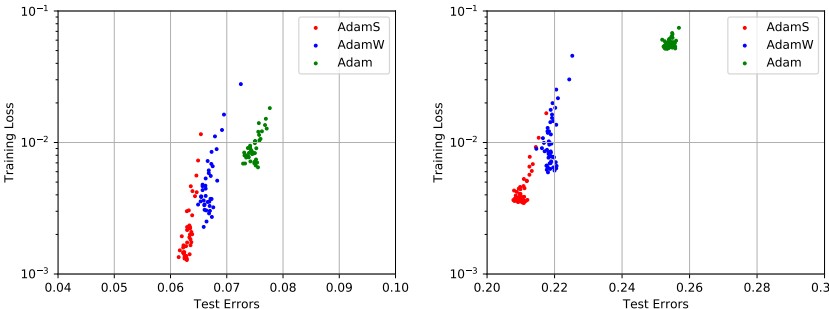

Figure 15: Even if with similar or higher training losses, AdamS still generalizes better than AdamW and Adam. The scatter plot of training losses and test errors during final 50 epochs of training VGG16 on CIFAR-10 and DenseNet121 on CIFAR-100.

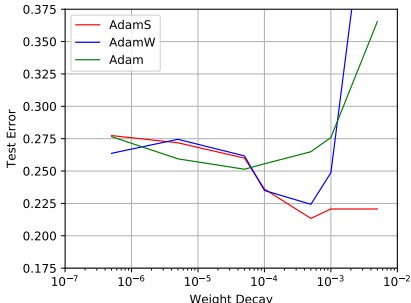

Figure 16: We compare the generalization of Adam, AdamW, and AdamS with various weight decay rates by training ResNet34 on CIFAR-100. The displayed weight decay of AdamW in the figure has been rescaled by the factor $= 0.001$. The optimal test performance of AdamS is significantly better than AdamW and Adam.

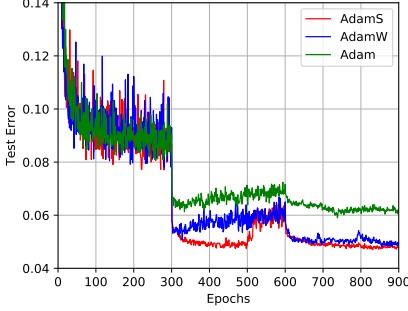

Figure 17: We train ResNet18 on CIFAR-10 for 900 epochs to explore the performance limit of AdamS, AdamW, and Adam. The learning rate is divided by 10 at the epoch of 300 and 600. AdamS achieves the most optimal test error, $4.70\%$.

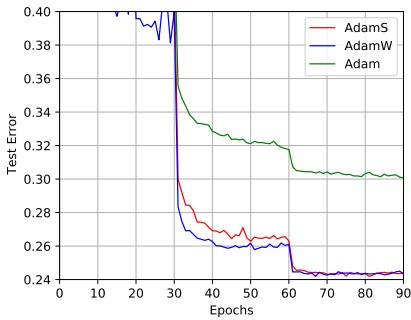

Figure 18: ResNet50 on ImageNet. The lowest Top-1 test errors of AdamS, AdamW, and Adam are 24.19%, 24.29%, and 30.07%, respectively.

Table 3: In the experiment of ResNet18 trained via SGD on CIFAR-10, we verified that the optimal weight decay is approximately inverse to the number of epochs. The predicted optimal weight decay is approximately $0.1 \times \text{Epochs}^{-1}$, because the optimal weight decay is $\lambda = 0.0005$ selected from $\{10^{-2}, 5 \times 10^{-3}, 10^{-3}, 5 \times 10^{-4}, 10^{-4}, 5 \times 10^{-5}, 10^{-5}, 5 \times 10^{-6}, 10^{-6}\}$ with 200 epochs as the base case. The observed optimal weight decay is selected from $\{\text{Epochs}^{-1}, 0.1 \times \text{Epochs}^{-1}, 0.01 \times \text{Epochs}^{-1}\}$. We observed that the optimal test errors are all corresponding to the predicted optimal weight decay $\lambda = 0.1 \times \text{Epochs}^{-1}$. At least in the sense of the order of magnitude, the predicted optimal weight decay is fully consistent with the observed optimal weight decay. Thus, the empirical results supports that the optimal weight decay is approximately inverse to the number of epochs in the common range of the number of epochs.

| EPOCHS | $\lambda = \text{Epochs}^{-1}$ | $\lambda = 0.1 \times \text{Epochs}^{-1}$ | $\lambda = 0.01 \times \text{Epochs}^{-1}$ |
|---|---|---|---|
| 50 | 74.06 | **7.12** | 7.50 |
| 100 | 22.04 | **5.56** | 6.01 |
| 200 | 11.81 | **5.02** | 5.61 |
| 1000 | 4.67 | **4.43** | 6.02 |
| 2000 | 4.59 | **4.48** | 5.70 |

We train ResNet18 on CIFAR-10 for 900 epochs to explore the performance limit of AdamS, AdamW, and Adam in Figure 17.

Figure 18 in Appendix shows that, AdamS can make marginal improvements over AdamW.

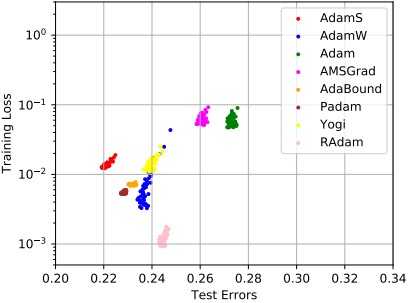

Figure 19: The scatter plot of training losses and test errors during final 40 epochs of training ResNet34 on CIFAR-100. Even with similar or higher training losses, AdamS still generalizes better than other Adam variants. We leave the scatter plot on CIFAR-10 in Appendix C.

## D   Additional Algorithms

We note that the implementation of AMSGrad in Algorithm 3 is the popular implementation in PyTorch. We use the PyTorch implementation in our paper, as it is widely used in practice.

---

**Algorithm 3:** AMSGrad/AMSGradW

$g_t = \nabla L(\theta_{t-1}) + \lambda\theta_{t-1}$;
$m_t = \beta_1 m_{t-1} + (1-\beta_1)g_t$;
$v_t = \beta_2 v_{t-1} + (1-\beta_2)g_t^2$;
$\hat{m}_t = \frac{m_t}{1-\beta_1^t}$;
$v_{max} = max(v_t, v_{max})$;
$\hat{v}_t = \frac{v_{max}}{1-\beta_2^t}$;
$\theta_t = \theta_{t-1} - \frac{\eta}{\sqrt{\hat{v}_t}+\epsilon}\hat{m}_t - \eta\lambda\theta_{t-1}$;

---

**Algorithm 4:** AMSGradS

$g_t = \nabla L(\theta_{t-1})$;
$m_t = \beta_1 m_{t-1} + (1-\beta_1)g_t$;
$v_t = \beta_2 v_{t-1} + (1-\beta_2)g_t^2$;
$\hat{m}_t = \frac{m_t}{1-\beta_1^t}$;
$v_{max} = max(v_t, v_{max})$;
$\hat{v}_t = \frac{v_{max}}{1-\beta_2^t}$;
$\bar{v}_t = mean(\hat{v}_t)$;
$\theta_t = \theta_{t-1} - \frac{\eta}{\sqrt{\hat{v}_t}+\epsilon}\hat{m}_t - \frac{\eta}{\sqrt{\bar{v}_t}}\lambda\theta_{t-1}$;

---

## E   Supplementary Experiments with Cosine Annealing Schedulers and Warm Restarts

In this section, we conducted comparative experiments on AdamS, AdamW, and Adam in the presence of cosine annealing schedulers and warm restarts proposed by Loshchilov and Hutter [31]. We set the learning rate scheduler with a recommended setting of Loshchilov and Hutter [31]: $T_0 = 14$ and $T_{mul} = 2$. The number of total epochs is 210. Thus, we trained each deep network for four runs of warm restarts, where the four runs have 14, 28, 56, and 112 epochs, respectively. Other hyperparameters and details are displayed in Appendix B.

We conducted comparative experiments on AdamS, AdamW, and Adam in the presence of cosine annealing schedulers and warm restarts proposed by Loshchilov and Hutter [31]. We set the learning rate scheduler with a recommended setting of Loshchilov and Hutter [31] Our experimental results in Figures 12 and 20 suggest that AdamS consistently outperforms AdamW and Adam in the presence of cosine annealing schedulers and warm restarts. It demonstrates that, with various learning rate schedulers, the advantage of SWD may generally hold.

Moreover, we did not empirically observe that cosine annealing schedulers with warm restarts may consistently outperform the common piecewise-constant learning rate schedulers for adaptive gradient methods. We noticed that Loshchilov and Hutter [31] empirically compared four-staged piecewise-constant learning rate schedulers with cosine annealing schedulers with warm restarts, and argue that cosine annealing schedulers with warm restarts are better. There may be two possible causes. First, three-staged piecewise-constant learning rate schedulers, which usually have a longer first stage and decay learning rates by multiplying $0.1$, are the recommended settings, while the four-staged piecewise-constant learning rate schedulers in Loshchilov and Hutter [31] are usually not optimal. Second, warm restarts may be helpful, while cosine annealing may be not. The ablation study on piecewise-constant learning rate schedulers with warm restarts is lacked. We argued that how to choose learning rate schedulers may still be an open question, considering the complex choices of schedulers and the complex loss landscapes.

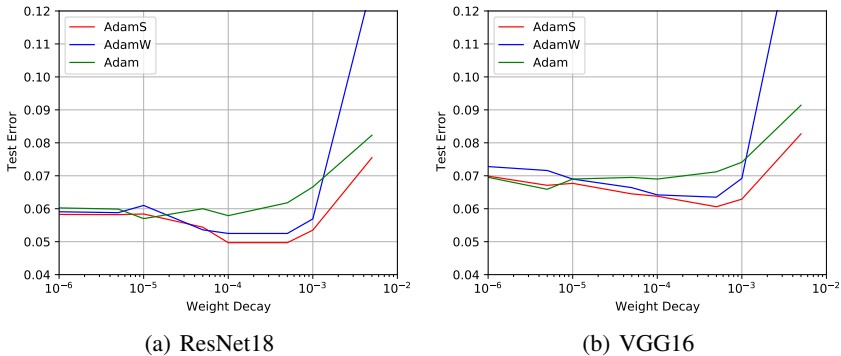

(a) ResNet18           (b) VGG16

Figure 20: The test errors of ResNet18 and VGG16 on CIFAR-10 under various weight decay with cosine annealing and warm restart schedulers. AdamS yields significantly better optimal test performance than AdamW and Adam.

