# OpenReview forum: "On the Overlooked Pitfalls of Weight Decay and How to Mitigate Them: A Gradient-Norm Perspective"
_NeurIPS.cc/2023/Conference — NeurIPS 2023 poster_

### Official Review · Reviewer_tfGK · 2023-07-06

**Soundness:** 3 good
**Presentation:** 4 excellent
**Contribution:** 3 good
**Rating:** 7
**Confidence:** 4

**Summary:**

The authors propose Scheduled Weight Decay (SWD), a method that mitigates the large gradient norm issue caused by constant weight decay factors. The authors demonstrate that SWD can improve the generalization performance of Adam and outperform other adaptive optimizers on CIFAR-10/100 datasets.

**Strengths:**

1. The proposed method is simple and easy to implement. Moreover, it addresses the long-standing generalization gap between adaptive optimizers and SGD on certain tasks.
2. The authors provide a theoretical analysis of the problem of unstable stationary points.
3. The authors conduct extensive experiments to validate their claims and compare their method with other optimizers.

**Weaknesses:**

1. The gradient analyzed in Theorem 2 seems to include the gradient of L2 regularization, which is not included in the empirical analysis (e.g. Fig. 2).
2. Table 1 in appendix and table 1 in the main text show very different results, especially for SGD. The reason for this difference should be made more clear.

**Questions:**

See above.

**Limitations:**

The authors have discussed the limitations of the proposed method on datasets other than CIFAR-10/100.

---

> ### Author Rebuttal · Authors · 2023-08-07
>
> We highly appreciate Reviewer tfGK’s kind support and helpful comments.
>
> The reviewer definitely realized the important value of identifying and mitigating the overlooked but serious pitfalls of weight decay. We gratefully hope the reviewer can express and insist on your opinion to avoid a possible loss of our community.
>
> We also properly addressed your concerns as follows.
>
> Q1: The gradient analyzed in Theorem 2 seems to include the gradient of L2 regularization, which is not included in the empirical analysis (e.g. Fig. 2).
>
> A1: Thanks for the helpful comment. We will discuss this point in the revision. Figure 2 indeed plots the gradient norms without regularization included. However, we argue that the empirical conclusions are same with or without regularization. because the numerical difference due to regularization is multiple orders of magnitude smaller than their gradient norms in our experiments.
>
>
> Q2: Table 1 in appendix and table 1 in the main text show very different results, especially for SGD. The reason for this difference should be made more clear.
>
> A2: Again, thanks for the helpful comment. The difference between two Tables 1 in the main text and the appendix lies in the hyperparameter choice of weight decay. We discuss why some previous papers’ results are more closed to Table 1 in the appendix. Some previous papers designed novel adaptive optimizers and claimed they generalize as well as SGD only because an improper weight decay strength is chosen for SGD. We have discussed this point in Line 85-101 of the appendix. We will follow your suggestion to present the point more clearly.

---

### Official Review · Reviewer_3dfT · 2023-07-07

**Soundness:** 3 good
**Presentation:** 2 fair
**Contribution:** 3 good
**Rating:** 6
**Confidence:** 3

**Summary:**

This paper studies the overlooked pitfalls of weight decay, a regularization technique used in deep neural networks (DNNs). The authors discovered that weight decay can lead to large gradient norms, particularly at the final phase of training, often indicating poor convergence and generalization. To address this issue, the authors propose a novel method called Scheduled Weight Decay (SWD), which dynamically adjusts weight decay strength according to the gradient norm and penalizes large gradient norms during training. The paper concludes that the SWD approach outperforms the conventional constant weight decay strategy, especially for the Adaptive Moment Estimation (Adam) optimization algorithm.

**Strengths:**

1. SWD dynamically adjusts weight decay, which can help to avoid large gradient norms that can lead to poor convergence and generalization. The paper demonstrates that SWD outperforms the traditional constant weight decay, especially in Adam optimization, potentially leading to better model performance.
2. The concept is simple to understand and can be easily implemented in most deep learning frameworks.

**Weaknesses:**

1. The proposal is specific to weight decay regularization and might not generalize to other regularization techniques.
2. SWD's performance might depend heavily on how well the scheduling function is chosen, which can be a complex task.
3. The paper doesn't discuss how SWD might affect the training time, and adjusting weight decay dynamically could potentially increase computational complexity.

**Questions:**

How does SWD affect the training time, given the additional complexity introduced by dynamically adjusting the weight decay?

**Limitations:**

Not applicable.

---

> ### Author Rebuttal · Authors · 2023-08-07
>
> We highly appreciate Reviewer 3dfT’s kind support and helpful comments.
>
> We gratefully hope the reviewer can express and insist on your opinions, which may help our community understand and employ weight decay better via our work.
>
> We also properly respond to your concerns as follows.
>
> Q1: The proposal is specific to weight decay regularization and might not generalize to other regularization techniques.
>
> A1: Thanks for the comment. It is true, while the value of our method is still significant given the importance and popularity of weight decay.
>
> Q2: SWD's performance might depend heavily on how well the scheduling function is chosen, which can be a complex task.
>
> A2: We respectfully note that SWD itself is an adaptive weight decay scheduler which directly depends on gradient norms and requires no further manual design. We also agree that it will be possible to design other more complex scheduler for weight decay in future.
>
> Q3: How does SWD affect the training time, given the additional complexity introduced by dynamically adjusting the weight decay?
>
> A3: In Line 201, we mentioned that the cost difference of AdamS versus AdamW/Adam is nearly ignorable (usually less than 5%) in practice. This is not surprising because computing mean(v) is very cheap compared with computing gradients per interation. We will emphasize this point more clearly in the revision.

---

> > ### Comment · Reviewer_3dfT · 2023-08-19
> >
> > After reading the authors response, I will keep the original score.

---

### Official Review · Reviewer_xE1B · 2023-07-31

**Soundness:** 2 fair
**Presentation:** 2 fair
**Contribution:** 1 poor
**Rating:** 4
**Confidence:** 4

**Summary:**

This paper studies the role of weight decay and its connection with large gradient norms in deep learning settings. In particular, the paper highlights differences in variants of weight decay and also the effect of weight decay on gradient norm in the final phase. Based on the observation that weight decay yields large gradient norms, the authors propose a scheduler for weight decay, called the Scheduled Weight Decay, which dynamically adjusts the weight decay strength.

**Strengths:**

The paper is reasonably well-written and the motivation for studying the effect of weight decay is both important and clear. The approach is easy to implement and the results on some smaller scale benchmarks seem encouraging.

**Weaknesses:**

(1) The theoretical results are weak and not very interesting. Theorem 2 follows from standard convergence rates for SGD. I could not find any novelty in this result. The authors should highlight the novelty of the result and proof technique in the main paper.

(2) While Theorem 1 seems interesting, it seems somewhat weak. The lower bound on the difference in learning rate is somewhat artificial and unreasonable. For instance, consider decreasing learning rates of SGD typically used in machine learning settings i.e., eta_t >= eta_{t+1}. Then Theorem 1 assumes delta <= eta_t - eta_{t+1}. This implies eta_0 >= delta . t for any t > 0. This makes sense if delta is going to 0 as t -> infinity. Maybe I am missing something but would really appreciate it if the authors can comment about this.

(3) Theorem 1 & 2 seem to analyze gradients of two different functions, which doesn't seem like a proper comparison. More remarks regarding this will be valuable. Furthermore, I would like to get clarification if the y-axis in plots of Figure 2 are gradient norms with regularization included.

(4) Definition 1 is very unclear. The authors should provide more remarks around definition 1 to help readers understand the definition better.

(5) The dependence of C_2 on sup norm of theta is somewhat weird in Theorem 2. Are you assuming ||theta|| is bounded? (otherwise rewrite the theorem statement without this dependence) .

(6) I think the theoretical basis for Gradient-Norm-Aware Scheduled Weight Decay is weak as presented in the paper. While I understand the intuition, it is not clear why this particular proposal is meaningful. Since the theoretical basis for this is lacking, I would expect much more comprehensive experiments to support the method. Unfortunately, the empirical analysis in the paper is somewhat limited.

Line 20: eta -> eta_t is the learning rate & Line 125: E[||∇L(θ, X) − ∇L(θ)||^2] ≤ \sigma^2?


Post-rebuttal
==========
I think the theoretical analysis is still not convincing but given that these observations are interesting, I am slightly increasing the score.

**Questions:**

Please refer to the concerns raised in the weakness section.

**Limitations:**

Not applicable.

---

> ### Author Rebuttal · Authors · 2023-08-07
>
> We sincerely appreciate Reviewer xE1B’s hard work and helpful comments.
>
> The comments shows that the reviewer only has concerns about theoretical evidences in our work. Our theoretical analysis is proposed not as a main contribution but to explain our interesting findings.
>
> Our main contributions are revealing the overlooked pitfalls of weight decay and how to mitigate them. If a simple theoretical mechanism can well describe the pitfalls, it will be an advantage rather than a weakness. Given the importance of weight decay, our contributions are significant.
>
> If the reviewer indeed accepts the reported overlooked pitfalls and the effectiveness of SWD, we strongly encourage the reviewer to re-evaluate the importance of our contributions.
>
> We also try to address the mentioned weaknesses about theoretical evidences as follows.
>
> Q1: Theorem 2 follows from standard convergence rates for SGD. I could not find any novelty in this result. The authors should highlight the novelty of the result and proof technique in the main paper.
>
> A1: We frankly admit that Theorem 2 can be easily obtained from standard convergence rates with the modification of weight decay. However, we believe that the simplicity of theoretical analysis is not a weakness. Again, our theoretical analysis is proposed not as a main contribution but to explain our interesting findings, namely the large-gradient-norm pitfall and SWD. We argue with Ockham's Razor that if a simple theory can explain an interesting novel finding, it will be unnecessary to pursue a complex theory with more tricks.
>
> Q2: While Theorem 1 seems interesting, it seems somewhat weak. The lower bound on the difference in learning rate is somewhat artificial and unreasonable. For instance, consider decreasing learning rates of SGD typically used in machine learning settings i.e., eta_t >= eta_{t+1}. Then Theorem 1 assumes delta <= eta_t - eta_{t+1}. This implies eta_0 >= delta . t for any t > 0. This makes sense if delta is going to 0 as t -> infinity. Maybe I am missing something but would really appreciate it if the authors can comment about this.
>
> A2: We would sincerely apologize if we misunderstood your question. In principle, if t approaches infinity and $\eta_{t}$ approaches, $delta$ and the lower bound indeed will approach to zero. However, in practice with finite iterations and a learning rate schedule, people can only have a finite learning rate. More importantly, Theorem 1 suggests that adaptive optimizaters can lead to a more significant large-gradient-norm pitfall, as $\delta$ as well as its preconditioned learning rate $\frac{\eta_{t}}{\sqrt{v_{t}}}$ can be unstable and large during training.
>
> Q3: Theorem 1 & 2 seem to analyze gradients of two different functions, which doesn't seem like a proper comparison. More remarks regarding this will be valuable. Furthermore, I would like to get clarification if the y-axis in plots of Figure 2 are gradient norms with regularization included.
>
> A3: Thanks for the constructive comment. Theorem 1 and 2 both analyze the regularized loss function $f(\theta)$, so the messages carried by them are consistent. We note that the expression of $f(\theta)$ can be slightly different due to the types of weight decay. We use the form of vanilla weight decay in Theorem 1 and use the form standard $L_{2}$ regularization/decoupled weight decay in Theorem 2, where $L_{2}$ regularization and decoupled weight decay are identical and common for SGD. We will follow your suggestion to make them more clear in the revision.
>
> Figure 2 plots the gradient norms without regularization included. However, the empirical conclusions are same with or without regularization, because the numerical difference due to regularization is multiple orders of magnitude smaller than their gradient norms in our experiments.
>
> Q4: Definition 1 is very unclear. The authors should provide more remarks around definition 1 to help readers understand the definition better.
>
> A4: Thanks for the constructive comment. We will provide more remarks and propose a more formal Definition 1.
>
> Q5: I think the theoretical basis for Gradient-Norm-Aware Scheduled Weight Decay is weak as presented in the paper. While I understand the intuition, it is not clear why this particular proposal is meaningful. Since the theoretical basis for this is lacking, I would expect much more comprehensive experiments to support the method. Unfortunately, the empirical analysis in the paper is somewhat limited.
>
> A5: We respectfully argue that the prior purpose of our theoretical analysis is to demonstrate the existence of the overlooked pitfalls of weight decay, because identifying the overlooked large-gradient-norm pitfall of weight decay is our first and most valuable contribution. After the demonstration of the overlooked pitfalls, the proposed Scheduled Weight Decay (SWD) method is a naturally inspired algorithm for mitigating the large-gradient-norm pitfall. The effectiveness of SWD for mitigating the overlooked pitfalls are extremely significant (e.g., Figures 3 and 4). We admit that we have no rigous generalization bound theory of scheduling weight decay at present. Formal theories of analyzing weight decay and learning rate schedulers are interesting. We will leave developing formal theories of analyzing weight decay schedulers as future explorations.
>
> Q6: Typos.
>
> A6: Thank a lot for pointing out the typos. We will correct them in the revision.

---

> > ### Author Response · Authors · 2023-08-14
> > **More Discussion?**
> >
> > We hope our responses could address your concerns.
> >
> > If there are any further questions, we are very glad to continue the discussion!

---

> > > ### Comment · Reviewer_xE1B · 2023-08-14
> > >
> > > Thanks for responding to the feedback.
> > >
> > > Regarding A1: Thanks for admitting this. Indeed, my concern is not regarding the simplicity of the result. The main concern is that result is somewhat trivial and does not need to stated as a primary result of the paper (citing a relevant work in this context and stating the result makes more sense).
> > >
> > > Regarding A2: Thanks for the additional context. In that case, it will be useful to subscript delta with at least t rather than present it as a constant? But the argument still feels unconvincing. If the lower bound essentially tends to 0 as t tends to infinity, may be I am missing something, is there a reason to mention it as "non-convergence". Also, one could also derive lower bounds for the setting considered in Theorem 2 as well? It is hard for the reader to extract the quantitative message of the result in Theorem 2 using an upper bound?
> > >
> > > Regarding A5: Thanks for the explanation. I respectfully disagree with the response. While the observation is interesting, I don't see a strong theoretical basis in the paper to motivate the approach. Without this, the paper seems somewhat incomplete and heuristic.
> > >
> > > I also didn't see any comment about about my question "The dependence of C_2 on sup norm of theta is somewhat weird in Theorem 2. Are you assuming ||theta|| is bounded? (otherwise rewrite the theorem statement without this dependence)".
> > >
> > > Overall, while I appreciate the author's response, my primary concerns still remain. I will keep my score unless a more convincing reasoning is provided.

---

> > > > ### Author Response · Authors · 2023-08-15
> > > > **Thanks for the feedbacks.**
> > > >
> > > > We sincerely thank the reviewer for the feedbacks.
> > > >
> > > > Regarding A1: Thanks for admitting this. Indeed, my concern is not regarding the simplicity of the result. The main concern is that result is somewhat trivial and does not need to stated as a primary result of the paper (citing a relevant work in this context and stating the result makes more sense).
> > > > Supplementary A1: Thanks for the suggestion. We will cite the relevant work and explain our results can be obtained from (but have a different form of) classical results with the modifications of the weight decay term.
> > > >
> > > > Regarding A2: Thanks for the additional context. In that case, it will be useful to subscript delta with at least t rather than present it as a constant? But the argument still feels unconvincing. If the lower bound essentially tends to 0 as t tends to infinity, may be I am missing something, is there a reason to mention it as "non-convergence". Also, one could also derive lower bounds for the setting considered in Theorem 2 as well? It is hard for the reader to extract the quantitative message of the result in Theorem 2 using an upper bound?
> > > >
> > > > Supplementary A2: We note that convergence require the condition which is often unsatisfied in practice, particularly when Adaptive Learning Rate is used. Our results reveal that non-convergence indeed happens and positively corelates to weight decay strength.
> > > > Theorem 2 shows that the upper bound of the squared gradient norm can converge to zero, while the convergence rate can be slow by large weight decay. As convergence analysis often study the norm upper bound, we think the upper bound is more informative here. Theorem 1 studies non-convergence case, so the lower bound is more informative there. We will clarify the messages conveyed by two thereoms more clearly in the revision
> > > >
> > > > Regarding A5: Thanks for the explanation. I respectfully disagree with the response. While the observation is interesting, I don't see a strong theoretical basis in the paper to motivate the approach. Without this, the paper seems somewhat incomplete and heuristic.
> > > >
> > > > Supplementary A5: We try to present a simple and fundamental theoretical basis, which successfully inspired us to observe interesting findings and design novel methods. We will appreciate it, if the reviewer may consider that the theoretical basis has really motivated our work.
> > > >
> > > > Q7: "The dependence of C_2 on sup norm of theta is somewhat weird in Theorem 2. Are you assuming ||theta|| is bounded? (otherwise rewrite the theorem statement without this dependence)".
> > > >
> > > > A7: We apologize for missing this question before. Yes, we assume that the weight norm is upper bounded during training. The weight norm bound is not theoretically guaranteed, because we can initialize model weights with arbitrarily large norms. However, in practice, we can empirically observe that the weight norm cannot arbitrarily increase and has the same order of magnitude as the initialized weight norm during training. So we introduce the assumption of the bounded weight norm in Theorem 2.

---

### Official Review · Reviewer_e7t4 · 2023-08-01

**Soundness:** 3 good
**Presentation:** 3 good
**Contribution:** 3 good
**Rating:** 6
**Confidence:** 3

**Summary:**

I would divide this paper's contributions into two parts.  The first part is an algorithm (a variant of Adam) which the authors argue generalizes better than Adam/AdamW and is easier to tune.  The second part is the justification for the effectiveness of that algorithm.

**The algorithm itself** The proposed algorithm, AdamS, is similar to AdamW, except that the weight decay strength is divided by the mean of the current squared gradient EMA.  This is effectively a schedule for weight decay.  It has the effect of penalizing the overall weights less strongly during phases of training when the overall gradient is large, and penalizing the overall weights more strongly during phases of training when the overall gradient is small.   For example, Figure 3 shows gradient norm and weight decay strength during the training of ResNet-34 on CIFAR-10; after the first learning rate drop, squared gradient norm grows, which causes the weight decay strength to shrink; after the second learning rate drop, squared learning rate plummets, which causes weight decay to rise.

**Experimental evaluation** Table 1 and section 5 experimentally evaluate AdamS and argue that the algorithm is better generalizing / easier to tune than AdamW.

**Justification**: The authors justify their algorithm along the following lines (which I disagree with; see more below):

1. When the learning rate is potentially changing, gradient descent with weight decay has trouble converging to stationary points - in particular, the authors argue (Theorem 1) that in this setting, the gradient norm will be lower bounded near stationary points.  Therefore, the authors argue that weight decay causes the gradient to be large, especially at the end of training.
2.  The authors argue that large gradients are bad for generalization.
3. As a consequence of #1 and #2, AdamW-style weight decay is bad for generalization (since it causes large gradients near stationary points, and large gradients are bad for generalization)

The authors acknowledge that weight decay on networks with normalization layers has been shown in the literature to control the effective learning rate, but insist that their paper's mechanism is unrelated to this, and propose an entirely different mechanism.

---

**post-rebuttal update**

After discussion with the authors, I'm raising my score from 3 ("reject") to 6 ("weak accept").    Firstly, it's clear that my original review was wrong; I had argued that the proposed algorithm's benefits are probably due to the interaction between weight decay and normalization layers, but the authors have correctly pointed out that the same phenomena occur on networks without normalization layers, and I have verified this myself.   The reason for 6 as opposed to 7 is that I believe that the proposed theoretical explanation for why weight decay would cause large gradients is not the real explanation (reviewer xE1B also didn't like this theoretical portion of the paper).  That said, it's entirely plausible that weight decay causing large gradients is one of the *many* aspects of deep learning that is just theoretically unexplainable at the present time.  So, my issue with the submission is less that the paper fails to include a correct theoretical justification, and more that it includes a faulty theoretical justification; I think it is better to have no justification at all than to have a faulty justification.

That being said, on the positive side, it seems there is a good chance that the proposed algorithm is a worthwhile addition to the deep learning toolbox (though I am not an expert at evaluating this type of contribution).

**Strengths:**

I'm not an expert in experimentally evaluating the performance of neural network training algorithms, but from the experiments in section 5 it does seem plausible to me that AdamS is indeed better / easier to tune than AdamW.

**Weaknesses:**

While I can accept that the algorithm may be a good idea, I strongly disagree with the proposed justification given in the paper.  I believe the effectiveness of the algorithm is probably unrelated to the justification given in the paper, and is instead closely linked to the implicit effects of weight decay on effective learning rate when the network has normalization layers.

First, I would note that all of the experiments in the paper are on networks with normalization layers.  The authors argue in Figure 2 that VGG-16 is "not scale invariant" but the VGG-16 architecture does have many BatchNorm layers, even though it is not fully scale-invariant, so I believe that the literature on scale invariance is still quite related to what is going on with the VGG-16.

For networks with normalization layers, there is a clear mechanism by which weight decay causes large gradients.  If $L$ is the loss for a scale-invariant network, then $\nabla L(c \theta) = \frac{1}{c} \nabla L(\theta)$, i.e. scaling down the weights $(c < 1)$ will automatically scale up the gradients (see Lemma 1 here: http://www.offconvex.org/2020/04/24/ExpLR1/).  The large gradients have been reported to cause the 'effective learning rate' of training to be higher, which is thought to be good for generalization.  My intuition for the mechanism in this paper is: if the gradient norm is too large, our effective learning rate will be too large for the algorithm to converge, so we need to decrease weight decay so the gradient norms come back down; if the gradient norm is too small, our effective learning rate will be too small (which makes convergence slow and is bad for generalization) so we need to increase weight decay so that the gradient norm moves back up.

By contrast, the proposed justification doesn't really make sense to me.  Figure 2 shows that higher weight decay causes growth in the gradient norm _in the middle of training_, not at convergence.  This is much more easily explained by scale-invariance (for the VGG-16, the scale invariance of certain layers) than by Theorems 1 and 2.

Figure 5 shows that weight decay increases the eigenvalue spectrum at convergence.  I believe this is related to cross-entropy loss, and wouldn't happen with e.g. squared loss.  For cross-entropy loss, the Hessian is small when the margins are large.  Adding weight decay prevents the margins from becoming large and therefore the Hessian from shrinking.

**Questions:**

How do the authors respond to my critique above?

**Limitations:**

discussed above

---

> ### Author Rebuttal · Authors · 2023-08-07
>
> We sincerely appreciate Reviewer e7t4 for hard work and admitting the value of our contribution.
>
> We notice that Reviewer e7t4 currently tends to reject our work only because the reviewer believes an alternative scale-invariance mechanism of our experiments. However, we respectfully note that the justification of Reviewer e7t4 made a key factual mistake, which misled Reviewer e7t4’s rating on our work.
>
> Standard VGG-16 (in the original paper and our paper) actually has no BatchNorm layers, while sometimes the so-called VGG-16-bn (with BatchNorm) is used in other papers. Thus, as we mentioned in the paper, scale invariance itself cannot explain the observed pitfalls of weight decay. So the belief that SWD is unrelated to large gradient norms and the associated ``evidence” are both inconvincible.
>
> We have duly addressed your justifications as follows.
>
> Q1: I believe the effectiveness of the algorithm is probably unrelated to the justification given in the paper, and is instead closely linked to the implicit effects of weight decay on effective learning rate when the network has normalization layers.
>
> A1: The belief that SWD is not related to mitigating large gradient norms has no true evidence. In contrast, we showed that SWD mitigates the overlooked pitfalls of weight decay with empirical and theoretical evidence. We did not observe that SWD or the pitfalls behave qualitatively different on ResNet (with BatchNorm) or VGG(without BatchNorm).
>
> Q2: I would note that all of the experiments in the paper are on networks with normalization layers. The authors argue in Figure 2 that VGG-16 is "not scale invariant" but the VGG-16 architecture does have many BatchNorm layers, even though it is not fully scale-invariant, so I believe that the literature on scale invariance is still quite related to what is going on with the VGG-16.
>
> A2: We respectfully point out that the first judgement is wrong. Standard VGG which we used is indeed not scale invariant and has no BatchNorm layer.
>
> Q3: My intuition for the mechanism in this paper is: if the gradient norm is too large, our effective learning rate will be too large for the algorithm to converge, so we need to decrease weight decay so the gradient norms come back down; if the gradient norm is too small, our effective learning rate will be too small (which makes convergence slow and is bad for generalization) so we need to increase weight decay so that the gradient norm moves back up. By contrast, the proposed justification doesn't really make sense to me.
>
> A3: We totally agree with the intuition (except that it only connects to scale invariance). Moreover, this intuition is actually NOT contradicted to the large-gradient norm pitfall. The large-gradient-norm pitfall is exactly a signal of poor convergence and generalization at the final training phase. The proposed SWD indeed automatically adjusts the strength of weight decay and then successfully mitigate the large gradient norms as well as poor convergence/generalization.
>
> Q4: Figure 2 shows that higher weight decay causes growth in the gradient norm in the middle of training, not at convergence. This is much more easily explained by scale-invariance (for the VGG-16, the scale invariance of certain layers) than by Theorems 1 and 2.
>
> A4: Figure 2 shows that weight decay causes growth in the gradient norm in both the middle phase and final phase of training, while the gradient norm at the final phase is relatively lower. This can be explained by Theorems 1 and 2, but cannot by explained scale invariance only (the VGG-16 has no scale-invariant layer). Moreover, the gradient norm growth at the final phase is more interesting, because it closely relates to poor convergence and generalization.
>
>
> Q5: Figure 5 shows that weight decay increases the eigenvalue spectrum at convergence. I believe this is related to cross-entropy loss, and wouldn't happen with e.g. squared loss. For cross-entropy loss, the Hessian is small when the margins are large. Adding weight decay prevents the margins from becoming large and therefore the Hessian from shrinking.
>
> A5: Thanks for the comments. We just reported the interesting observation that the Hessian eigenspectrum and minima sharpness are significantly affected by SWD. Your comment may be right but does not indicate any weakness of our work.
>
> Finally, we sincerely thanks Reviewer e7t4 for the hard work and comments again.
>
> We strongly encourage the reviewer to re-evaluate our work without the distraction of misunderstanding and factual mistake.
>
> We appreciate it very much in advance.

---

> > ### Comment · Reviewer_e7t4 · 2023-08-10
> > **question**
> >
> > Hmm, ok, I'll have to think about this.  A question: on networks with no batch normalization, does SGD with weight decay also cause high gradient norm mid-training?  (That is, does mid-training gradient norm get higher as weight decay strength gets higher?)  Or is this just something that Adam does?

---

> > > ### Comment · Reviewer_e7t4 · 2023-08-10
> > > **code for VGG**
> > >
> > > Additionally, could you point me towards code for a VGG-16 sized for CIFAR-10 with no batch norm?

---

> > > ### Author Response · Authors · 2023-08-11
> > > **Grateful thanks and Response to Additional Questions**
> > >
> > > We gratefully thank the reviewer for the prompt reply and carefully re-evaluate our work.
> > >
> > > Your responsibility will be highly appreciated and is exactly what the whole community eagerly expects.
> > >
> > > We respond to your additional questions as follows.
> > >
> > > Q6: on networks with no batch normalization, does SGD with weight decay also cause high gradient norm mid-training? (That is, does mid-training gradient norm get higher as weight decay strength gets higher?) Or is this just something that Adam does?
> > >
> > > A6: Yes, SGD with weight decay also significantly cause high gradient norm during training. The observation is quite general. It does not only happen to Adam.
> > >
> > > Q7: Additionally, could you point me towards code for a VGG-16 sized for CIFAR-10 with no batch norm?
> > >
> > > A7: Yes, of course. The following github repo (https://github.com/kuangliu/pytorch-cifar/blob/master/models/vgg.py) also contains a very population implementation of VGG for CIFAR-10/100.
> > >
> > > Moreover, the torchvision.models.vgg16 is also a standard VGG-16 without BatchNorm (usually for ImageNet). The VGG-16 with BatchNorm refers to as torchvision.models.vgg16_bn.

---

> > > > ### Comment · Reviewer_e7t4 · 2023-08-11
> > > > **clarification**
> > > >
> > > > The kuangliu repo is the one that I've used in the past for CIFAR architectures, but just to be clear, the VGG in this that repo has BN layers.  If I want to run the net that you used in your experiments, do I just comment out the BN layers?

---

> > > > > ### Author Response · Authors · 2023-08-12
> > > > > **More evidence. Ablation study on VGG16 with/without BatchNorm.**
> > > > >
> > > > > Q8: Clarification.
> > > > >
> > > > > A8: Yes, we may simply comment out the BN layers to obtain standard VGG16 with BatchNorm.
> > > > >
> > > > > We apologize for the confusion. In Figure 2, our intention was actually to specifically study a VGG16 without BatchNorm, as the caption indicates. The current writing may be not clear enough. We will carefully clarify how VGG16 with or without BatchNorm is chosen in the revision.
> > > > >
> > > > > Q10: Ablation Study. (While the reviewer did not directly request, we think the ablation study on BatchNorm can be very helpful.)
> > > > >
> > > > > A10: Inspired by the reviewer's interest in how the studied large-gradient-norm pitfalls of weight decay depend on scale invariance resulted by BatchNorm, we decide to add the ablation study on BatchNorm and show that our conclusion generally hold with or without BatchNorm.
> > > > >
> > > > > We think the ablation study will be helpful in the revision and can demonstrate that BatchNorm does not weaken our conclusions.
> > > > >
> > > > >
> > > > > We follow the setting in Figure 2, training both VGG16 (without BatchNorm) and VGG16BN (with BatchNorm) via AdamW.
> > > > >
> > > > > For VGG16, when we multiply the weight decay strength by 5, the gradient norm and the squared gradient norm will grow to $\times 33$ and $\times 26$, respectively.
> > > > >
> > > > > For VGG16BN, when we multiply the weight decay strength by 5, the gradient norm and the squared gradient norm will only grow to $\times 14$ and $\times 8$, respectively.
> > > > >
> > > > > (BTW, the gradient norm of VGG16BN is consistently smaller than the gradient norm of VGG16.)
> > > > >
> > > > >
> > > > > The ablation study results directly oppose the reviewer’s comment that “for networks with normalization layers, there is a clear mechanism by which weight decay causes large gradients...”.
> > > > >
> > > > > This study demonstrate that the large-gradient-norm pitfalls of weight decay is not resulted by BatchNorm. Contradicted to the reviewer’s mechanism, BatchNorm can even effectively mitigate the large-gradient-norm pitfalls of weight decay, if weight decay increases.
> > > > >
> > > > > We will present more ablation study results on normlization layers (e.g., using ResNet18 without BatchNorm) to demonstrate that scale invariance cannot explain our reported contributions.
> > > > >
> > > > > Finally, we sincerely thank the reviewer’s idea. It definitely inspires us to further improve our work and clarification for more readers.

---

> > > > > > ### Comment · Reviewer_e7t4 · 2023-08-14
> > > > > > **reproduced results**
> > > > > >
> > > > > > (Please ignore my now-deleted comment titled "trying to reproduce results")
> > > > > >
> > > > > > OK, I've successfully reproduced on a VGG-16 your finding that weight decay causes large gradients. I'm still not totally satisfied with the explanation for this effect that is given in the submission, so I'm going to poke around a little bit to see if I can understand what is going on.

---

> > > > > > > ### Author Response · Authors · 2023-08-14
> > > > > > > **Thanks for successfully reproducing the results**
> > > > > > >
> > > > > > >
> > > > > > > We sincerely appreciate the reviewer's effort for reproducing the results and verifying that the large-gradient-norm pitfall of weight decay indeed happens to VGG with or without BatchNorm.
> > > > > > >
> > > > > > > Q11: I'm still not totally satisfied with the explanation for this effect that is given in the submission, so I'm going to poke around a little bit to see if I can understand what is going on.
> > > > > > >
> > > > > > > A11: Even if it is possible to develop other theoretical mechanisms for the novel and interesting discovery in our work, the possibility of other theoretical mechanisms does not make our theoretical analysis wrong or meaningless. Moreover, in this paper, we made contributions beyond just theoretical explanations. We will appreciate it if the reviewer may also think about the novelty and importance of our discovery itself.

---

### Decision · Program_Chairs · 2023-09-21

**Decision:**

Accept (poster)

**Comment:**

This paper looks at pitfalls of current approaches using weight decay. The theory for this paper is a little dubious and is not rigorous enough (see for instance review of Reviewer xE1B). Furthermore, algorithm is not well motivated. However, I think the observations are interesting and are worth being carefully examined by the community. For this reason, I recommend acceptance contingent on the commitment that the theoretical analysis and motivation will be carefully examined and discussed based on xE1B's review.